# Conflict-Averse Gradient Descent
# for Multi-task Learning

†**Bo Liu,** †**Xingchao Liu,** ‡**Xiaojie Jin,** †,§**Peter Stone,** †**Qiang Liu**
†The University of Texas at Austin, §Sony AI, ‡Bytedance Research
{bliu,xcliu,pstone,lqiang}@cs.utexas.edu, xjjin0731@gmail.com

## Abstract

The goal of multi-task learning is to enable more efficient learning than single task learning by sharing model structures for a diverse set of tasks. A standard multi-task learning objective is to minimize the average loss across all tasks. While straightforward, using this objective often results in much worse final performance for each task than learning them independently. A major challenge in optimizing a multi-task model is the *conflicting gradients*, where gradients of different task objectives are not well aligned so that following the average gradient direction can be detrimental to specific tasks' performance. Previous work has proposed several heuristics to manipulate the task gradients for mitigating this problem. But most of them lack convergence guarantee and/or could converge to any Pareto-stationary point. In this paper, we introduce Conflict-Averse Gradient descent (CAGrad) which minimizes the average loss function, while leveraging the worst local improvement of individual tasks to regularize the algorithm trajectory. CAGrad balances the objectives automatically and still provably converges to a minimum over the average loss. It includes the regular gradient descent (GD) and the multiple gradient descent algorithm (MGDA) in the multi-objective optimization (MOO) literature as special cases. On a series of challenging multi-task supervised learning and reinforcement learning tasks, CAGrad achieves improved performance over prior state-of-the-art multi-objective gradient manipulation methods. Code is available at `https://github.com/Cranial-XIX/CAGrad`.

## 1 Introduction

Multi-task learning (MTL) refers to learning a single model that can tackle multiple different tasks [11, 28, 44, 38]. By sharing parameters across tasks, MTL methods learn more efficiently with an overall smaller model size compared to learning with separate models [38, 40, 25]. Moreover, it has been shown that MTL could in principle improve the quality of the learned representation and therefore benefit individual tasks [35, 43, 34]. For example, an early MTL result by [2] demonstrated that training a neural network to recognize doors could be improved by simultaneously training it to recognize doorknobs.

However, learning multiple tasks simultaneously can be a challenging optimization problem because it involves multiple objectives [38]. The most popular MTL objective in practice is the average loss over all tasks. Even when this average loss is exactly the true objective (as opposed to only caring about a single task as in the door/doorknob example), directly optimizing the average loss could lead to undesirable performance, e.g. the optimizer struggles to make progress so the learning performance significantly deteriorates. A known cause of this phenomenon is the *conflicting gradients* [41]: gradients from different tasks 1) may have varying scales with the largest gradient dominating the update, and 2) may point in different directions so that directly optimizing the average loss can be quite detrimental to a specific task's performance.

35th Conference on Neural Information Processing Systems (NeurIPS 2021).

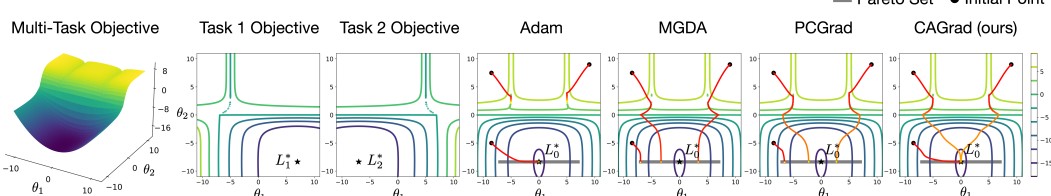

Figure 1: The optimization challenges faced by gradient descent (GD) and existing gradient manipulation methods like the multiple gradient descent algorithm (MGDA) [6] and PCGrad [41]. MGDA, PCGrad and CAGrad are applied on top of the Adam optimizer [16]. For each methods, we repeat 3 runs of optimization from different initial points (labeled with ●). Each optimization trajectory is colored from red to yellow. GD with Adam gets stuck on two of the initial points because the gradient of one task overshadows that of the other task, causing the algorithm to jump back and forth between the walls of a steep valley without making progress along the floor of the valley. MGDA and PCGrad stop optimization as soon as they reach the Pareto set.

To address this problem, previous work either adaptively re-weights the objectives of each task based on heuristics [3, 15] or seeks a better update vector [30, 41] by manipulating the task gradients. However, existing work often lacks convergence guarantees or only provably converges to any point on the Pareto set of the objectives. This means the final convergence point of these methods may largely depend on the initial model parameters. As a result, it is possible that these methods over-optimize one objective while overlooking the others (See Fig. 1).

Motivated by the limitation of current methods, we introduce Conflict-Averse Gradient descent (CAGrad), which reduces the conflict among gradients and still provably converges to a minimum of the average loss. The idea of CAGrad is simple: it looks for an update vector that maximizes the worst local improvement of any objective in a neighborhood of the average gradient. In this way, CAGrad automatically balances different objectives and smoothly converges to an optimal point of the average loss. Specifically, we show that vanilla gradient descent (GD) and the multiple gradient descent algorithm (MGDA) are special cases of CAGrad (See Sec. 3.1). We demonstrate that CAGrad can improve over prior state-of-the-art gradient manipulation methods on a series of challenging multi-task supervised, semi-supervised, and reinforcement learning problems.

## 2 Background

In this section, we first introduce the problem setup of multi-task learning (MTL). Then we analyze the optimization challenge of MTL and discuss the limitation of prior gradient manipulation methods.

### 2.1 Multi-task Learning and its Challenge

In multi-task learning (MTL), we are given $K \geq 2$ different tasks, each of which is associated with a loss function $L_i(\theta)$ for a shared set of parameters $\theta$. The goal is to find an optimal $\theta \in \mathbb{R}^m$ that achieves low losses across all tasks. In practice, a standard objective for MTL is minimizing the average loss over all tasks:

$$\theta^* = \arg\min_{\theta \in \mathbb{R}^m} \left\{ L_0(\theta) \triangleq \frac{1}{K} \sum_{i=1}^{K} L_i(\theta) \right\}. \tag{1}$$

Unfortunately, directly optimizing (1) using gradient descent may significantly compromise the optimization of individual losses in practice. A major source of this phenomenon is known as the conflicting gradients [41].

**Optimization Challenge: Conflicting Gradients**   Denote by $g_i = \nabla L_i(\theta)$ the gradient of task $i$, and $g_0 = \nabla L_0(\theta) = \frac{1}{K} \sum_i^K g_i$ the averaged gradient. With learning rate $\alpha \in \mathbb{R}^+$, $\theta \leftarrow \theta - \alpha g_0$ is the steepest descent update that appears to be the most natural update to follow when optimizing (1). However, $g_0$ may conflict with individual gradients, i.e. $\exists\ i,\ \langle g_i, g_0 \rangle < 0$. When this conflict is large, following $g_0$ will decrease the performance on task $i$. As observed by [41] and illustrated in Fig. 1, when $\theta$ is near a steep "valley", where a specific task's gradient dominates the update, manipulating the direction and magnitude of $g_0$ often leads to better optimization.

## 2.2 Prior Attempts and Convergence Issues

Several methods have been proposed to manipulate the task gradients to form a new update vector and have shown improved performance on MTL. Sener et al. apply the multiple-gradient descent algorithm (MGDA) [6] for MTL, which directly optimizes towards the Pareto set [30]. Chen et al. dynamically re-weight each $L_i$ using a pre-defined heuristic [3]. More recently, PCGrad identifies conflicting gradients as the motivation behind manipulating the gradients and projects each task gradient to the normal plane of others to reduce the conflict [41]. While all these methods have shown success at improving the learning performance of MTL, they manipulate the gradient without respecting the original objective (1). Therefore, these methods could in principle converge to any point in the Pareto set (See Fig. 1 and Sec. 3.2). We provide the detailed algorithms of MGDA and PCGrad in Appendix A.1 and A.2, and a visualization of the update vector by each method in Fig. 2.

## 3 Method

We introduce our main algorithm, Conflict-Averse Gradient descent in Sec. 3.1, and then show theoretical analysis in Sec. 3.2.

### 3.1 Conflict-Averse Gradient Descent

Assume we update $\theta$ by $\theta' \leftarrow \theta - \alpha d$, where $\alpha$ is a step size and $d$ an update vector. We want to choose $d$ to decrease not only the average loss $L_0$, but also every individual loss. To do so, we consider the minimum decrease rate across the losses,

$$R(\theta, d) = \max_{i \in [K]} \left\{ \frac{1}{\alpha} \left( L_i(\theta - \alpha d) - L_i(\theta) \right) \right\} \approx - \min_{i \in [K]} \langle g_i, d \rangle, \tag{2}$$

where we use the first-order Taylor approximation assuming $\alpha$ is small. If $R(\theta, d) < 0$, it means that all losses are decreased with the update given a sufficiently small $\alpha$. Therefore, $R(\theta, d)$ can be regarded as a measurement of conflict among objectives.

With the above measurement, our algorithm finds an update vector that minimizes such conflict to mitigate the optimization challenge while still converging to an optimum of the main objective $L_0(\theta)$. To this end, we introduce Conflict-Averse Gradient descent (CAGrad), which on each optimization step determines the update $d$ by solving the following optimization problem:

$$\max_{d \in \mathbb{R}^m} \min_{i \in [K]} \langle g_i, d \rangle \quad \text{s.t.} \quad \|d - g_0\| \le c \|g_0\|, \tag{3}$$

Here, $c \in [0, 1)$ is a pre-specified hyper-parameter that controls the convergence rate (See Sec. 3.2). The optimization problem (3) looks for the best update vector within a local ball centered at the averaged gradient $g_0$, which also minimizes the conflict in losses measured by (2). Since we focus on MTL and choose the average loss as the main objective, $g_0$ is the average gradient. However, CAGrad also applies when $g_0$ is the gradient of some other user-specified objective. We leave exploring this possibility as a future direction.

**Dual Objective** The optimization problem (3) involves decision variable $d$ that has the same dimension as the number of parameters in $\theta$, which could be millions for a deep neural network. It is not practical to directly solve for $d$ on every optimization step. However, the dual problem of Eq. (3), as we will derive in the following, only involves solving for a decision variable $w \in \mathbb{R}^K$, which can be efficiently found using standard optimization libraries [7]. Specifically, first note that $\min_i \langle g_i, d \rangle = \min_{w \in \mathcal{W}} \langle \sum_i w_i g_i, d \rangle$, where $w = (w_1, \ldots, w_K) \in \mathbb{R}^K$ and $\mathcal{W}$ denotes the probability simplex, i.e. $\mathcal{W} = \{w \colon \sum_i w_i = 1 \text{ and } w_i \ge 0\}$. Denote $g_w = \sum_i w_i g_i$ and $\phi = c^2 \|g_0\|^2$. The Lagrangian of the objective in Eq. (3) is

$$\max_{d \in \mathbb{R}^m} \min_{\lambda \ge 0, w \in \mathcal{W}} g_w^\top d - \lambda(\|g_0 - d\|^2 - \phi)/2.$$

Since the objective for $d$ is concave with linear constraints, by switching the min and max, we reach the dual form without changing the solution by Slater's condition:

$$\min_{\lambda \ge 0, w \in \mathcal{W}} \max_{d \in \mathbb{R}^m} g_w^\top d - \lambda \|g_0 - d\|^2 /2 + \lambda \phi/2.$$

---

**Algorithm 1** Conflict-averse Gradient Descent (CAGrad) for Multi-task Learning

---

**Input**: Initial model parameter vector $\theta_0$, differentiable loss functions $\{L_i\}_{i=1}^K$, a constant $c \in [0, 1)$ and learning rate $\alpha \in \mathbb{R}^+$.

**repeat**

At the $t$-th optimization step, define $g_0 = \frac{1}{K} \sum_{i=1}^K \nabla L_i(\theta_{t-1})$ and $\phi = c^2 \|g_0\|^2$. Solve

$$\min_{w \in \mathcal{W}} F(w) := g_w^\top g_0 + \sqrt{\phi} \|g_w\|, \text{ where } g_w = \frac{1}{K} \sum_{i=1}^K w_i \nabla L_i(\theta_{t-1}).$$

Update $\theta_t = \theta_{t-1} - \alpha \left( g_0 + \frac{\phi^{1/2}}{\|g_w\|} g_w \right)$.

**until** convergence

---

We end up with the following optimization problem w.r.t. $w$ after several steps of calculus,

$$w^* = \arg\min_{w \in \mathcal{W}} g_w^\top g_0 + \sqrt{\phi} \|g_w\|,$$

where the optimal $\lambda^* = \|g_{w^*}\| / \phi^{1/2}$ and the optimal update $d^* = g_0 + g_{w^*}/\lambda^*$. The detailed derivation is provided in Appendix A.3 and the entire CAGrad algorithm is summarized in Alg. 1. The dimension of $w$ equals to the number of objectives $K$, which usually ranges from 2 to tens and is much smaller than the number of parameters in a neural network. Therefore, in practice, we solve the dual objective to perform the update of CAGrad.

**Remark**  In Alg. 1, when $c = 0$, CAGrad recovers the typical gradient descent with $d = g_0$. On the other hand, when $c \to \infty$, then minimizing $F(w)$ is equivalent to $\min_w \|g_w\|$. This coincides with the multiple gradient descent algorithm (MGDA) [6], which uses the minimum norm vector in the convex hull of the individual gradients as the update direction (see Fig. 2; second column). MGDA is a gradient-based multi-objective optimization designed to converge to an arbitrary point on the Pareto set, that is, it leaves all the points on the Pareto set as fixed points (and hence can not control which specific point it will converge to). It is different from our method which targets to minimize $L_0$ while using gradient conflict to regularize the optimization trajectory. As we will analyze in the following section, to guarantee that CAGrad converges to an optimum of $L_0(\theta)$, we have to ensure $0 \le c < 1$.

### 3.2  Convergence Analysis

In this section we first formally introduce the related Pareto concepts and then analyze CAGrad's convergence property. Particularly, in Alg. 1, when $c < 1$, CAGrad is guaranteed to converge to a minimum point of the average loss $L_0$.

**Pareto Concepts**  Unlike single task learning where any two parameter vectors $\theta_1$ and $\theta_2$ can be ordered in the sense that either $L(\theta_1) \le L(\theta_2)$ or $L(\theta_1) \ge L(\theta_2)$ holds, MTL could have two parameter vectors where one performs better for task $i$ and the other performs better for task $j \ne i$. To this end, we need the notion of Pareto-optimality [13].

**Definition 3.1** (Pareto optimal and stationary points). *Let $\boldsymbol{L}(\theta) = \{L_i(\theta): i \in [K]\}$ be a set of differentiable loss functions from $\mathbb{R}^m$ to $\mathbb{R}$. For two points $\theta, \theta' \in \mathbb{R}^m$, we say that $\theta$ is Pareto dominated by $\theta'$, denoted by $\boldsymbol{L}(\theta') \prec \boldsymbol{L}(\theta)$, if $L_i(\theta') \le L_i(\theta)$ for all $i \in [K]$ and $\boldsymbol{L}(\theta') \ne \boldsymbol{L}(\theta)$. A point $\theta \in \mathbb{R}^m$ is said to be Pareto-optimal if there exists no $\theta' \in \mathbb{R}^m$ such that $\boldsymbol{L}(\theta') \prec \boldsymbol{L}(\theta)$. The set of all Pareto-optimal points is called the Pareto set. A point $\theta$ is called Pareto-stationary if we have $\min_{w \in \mathcal{W}} \|g_w(\theta)\| = 0$, where $g_w(\theta) = \sum_{i=1}^K w_i \nabla L_i(\theta)$, and $\mathcal{W}$ is the probability simplex on $[K]$.*

Similar to the case of single-objective differentiable optimization, a local Pareto optimal point $\theta$ must be Pareto stationary (see e.g., [6]).

**Theorem 3.2** (Convergence of CAGrad). *Assume the individual loss functions $L_0, L_1, \ldots, L_K$ are differentiable on $\mathbb{R}^m$ and their gradients $\nabla L_i(\theta)$ are all $H$-Lipschitz, i.e. $\|\nabla L_i(x) - \nabla L_i(y)\| \le H \|x - y\|$ for $i = 0, 1, \ldots, K$ where $0 \le H \le \infty$. Assume $L_0^* = \inf_{\theta \in \mathbb{R}^m} L_0(\theta) > -\infty$.*

*With a fixed step size $\alpha$ satisfying $0 < \alpha \le 1/H$, we have for the CAGrad in Alg. 1:*

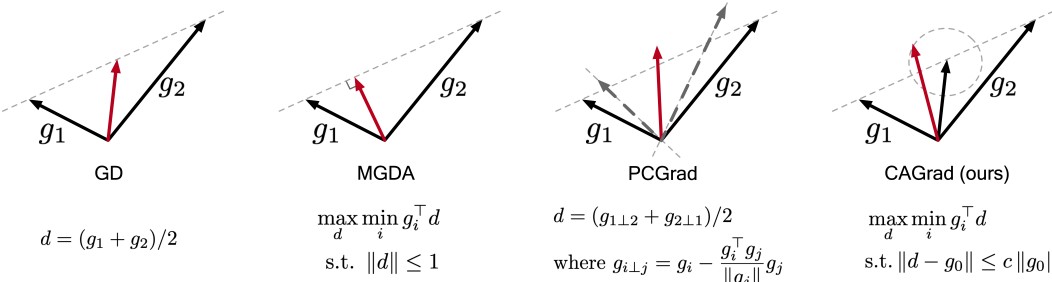

Figure 2: The combined update vector $d$ (in red) of a two-task learning problem with gradient descent (GD), multiple gradient descent algorithm (MGDA), PCGrad and Conflict-Averse Gradient descent (CAGrad). The two task-specific gradients are labeled $g_1$ and $g_2$. MGDA's objective is given in its primal form (See Appendix A.1). For PCGrad, each gradient is first projected onto the normal plane of the other (the dashed arrows). Then the final update vector is the average of the two projected gradients. CAGrad finds the best update vector within a ball around the average gradient that maximizes the worse local improvement between task 1 and task 2.

*1) For any $c \geq 1$, all the fixed points of CAGrad are Pareto-stationary points of $(L_0, L_1, \ldots, L_K)$.*

*2) In particular, if we take $0 \leq c < 1$, then CAGrad satisfies*

$$\sum_{t=0}^{T} \|\nabla L_0(\theta_t)\|^2 \leq \frac{2(L_0(\theta_0) - L_0^*)}{\alpha(1 - c^2)}.$$

This means that the algorithm converges to a stationary point of $\nabla L_0$ if we take $0 \leq c < 1$. The proof is in Appendix A.3. As we discuss earlier, unlike our method, MGDA is designed to converge to an arbitrary point on the Pareto set, without explicit control of which point it will converges to. Another algorithm with similar property is PCGrad [41], which is a gradient-based algorithm that mitigates the conflicting gradients problem by removing the conflicting components of each gradient with respect to the other gradients before averaging them to form the final update; see Fig. 2, third column for the illustration. Similar to MGDA, as shown in [41], PCGrad also converges to an arbitrary Pareto point without explicit control of which point it will arrive at.

### 3.3 Practical Speedup

A typical drawback of methods that manipulate gradients is the computation overhead. For computing the optimal update vector, a method usually requires $K$ back-propagations to find all individual gradients $g_i$, in addition to the time required for optimization. This can be prohibitive for the scenario with many tasks. To this end, we propose to only sample a subset of tasks $S \subseteq [K]$, compute their corresponding gradients $\{g_i \mid i \in S\}$ and the averaged gradient $g_0$. Then we optimize $d$ in:

$$\max_{d \in \mathbb{R}^m} \min \left( \left\langle \frac{Kg_0 - \sum_{i \in S} g_i}{K - |S|}, d \right\rangle, \quad \min_{i \in S} \langle g_i, d \rangle \right) \quad \text{s.t.} \quad \|d - g_0\| \leq c \|g_0\| \tag{4}$$

**Remark** Note that the convergence guarantee in Thm. 3.2 still holds for Eq. 4 as the constraint does not change (See Appendix A.3). The time complexity is $\mathcal{O}((|S|N + T)$, where $N$ denotes the time for one pass of back-propagation and $T$ denotes the optimization time. For few-task learning ($K < 10$), usually $T \ll N$. When $S = [K]$, we recover the full CAGrad algorithm.

## 4 Related Work

**Multi-task Learning** Due to its benefit with regards to data and computational efficiency, multi-task learning (MTL) has broad applications in vision, language, and robotics [11, 28, 22, 44, 38]. A number of MTL-friendly architectures have been proposed using task-specific modules [25, 11], attention-based mechanisms [21] or activating different paths along the deep networks to tackle MTL [27, 40]. Apart from designing new architectures, another branch of methods focus on decomposing a large problem into smaller local problems that could be quickly learned by smaller models [29, 26, 37, 8]. Then a unified policy is learned from the smaller models using knowledge distillation [12].

**MTL Optimization**  In this work, we focus on the optimization challenge of MTL [38]. Gradient manipulation methods are designed specifically to balance the learning of each task. The simplest form of gradient manipulation is to re-weight the task losses based on specific criteria, e.g., uncertainty [15], gradient norm [3], or difficulty [9]. These methods are mostly heuristics and their performance can be unstable. Recently, two methods [30, 41] that manipulate the gradients to find a better local update vector have become popular. Sener et al [30] view MTL as a multi-objective optimization problem, and use multiple gradient descent algorithm for optimization. PCGrad [41] identifies a major optimization challenge for MTL, the conflicting gradients, and proposes to project each task gradient to the normal plane of other task gradients before combining them together to form the final update vector. Though yielding good empirical performance, both methods can only guarantee convergence to a Pareto-stationary point, but not knowing where it exactly converges to. More recently, GradDrop [4] randomly drops out task gradients based on how much they conflict. IMTL-G [20] seeks an update vector that has equal projections on each task gradient. RotoGrad [14] separately scales and rotates task gradients to mitigate optimization conflict.

Our method, CAGrad, also manipulates the gradient to find a better optimization trajectory. Like other MTL optimization techniques, CAGrad is model-agnostic. However, unlike prior methods, CAGrad converges to the optimal point in theory and achieves better empirical performance on both toy multi-objective optimization tasks and real-world applications.

## 5   Experiment

We conduct experiments to answer the following questions:

**Question (1)** Do CAGrad, MGDA and PCGrad behave consistently with their theoretical properties in practice? (yes)

**Question (2)** Does CAGrad recover GD and MGDA when varying the constant $c$? (yes)

**Question (3)** How does CAGrad perform in both performance and computational efficiency compared to prior state-of-the-art methods, on challenging multi-task learning problems under the supervised, semi-supervised and reinforcement learning settings? (CAGrad improves over prior state-of-the-art methods under all settings)

### 5.1   Convergence and Ablation over c

To answer questions **(1)** and **(2)**, we create a toy optimization example to evaluate the convergence of CAGrad compared to MGDA and PCGrad. On the same toy example, we ablate over the constant $c$ and show that CAGrad recovers GD and MGDA with proper $c$ values. Next, to test CAGrad on more complicated neural models, we perform the same set of experiments on the Multi-Fashion+MNIST benchmark [19] with a shrinked LeNet architecture [18] (in which each layer has a reduced number of neurons compared to the original LeNet). Please refer to Appendix B for more details.

For the toy optimization example, we modify the toy example used by Yu et al. [41] and consider $\theta = (\theta_1, \theta_2) \in \mathbb{R}^2$ with the following individual loss functions:

$$L_1(\theta) = c_1(\theta)f_1(\theta) + c_2(\theta)g_1(\theta) \ \text{ and } \ L_2(\theta) = c_1(\theta)f_2(\theta) + c_2(\theta)g_2(\theta), \ \text{where}$$
$$f_1(\theta) = \log\big(\max(|0.5(-\theta_1 - 7) - \tanh(-\theta_2)|, \ 0.000005)\big) + 6,$$
$$f_2(\theta) = \log\big(\max(|0.5(-\theta_1 + 3) - \tanh(-\theta_2) + 2|, \ 0.000005)\big) + 6,$$
$$g_1(\theta) = \big((-\theta_1 + 7)^2 + 0.1 * (-\theta_2 - 8)^2\big)/10 - 20,$$
$$g_2(\theta) = \big((-\theta_1 - 7)^2 + 0.1 * (-\theta_2 - 8)^2\big)/10 - 20,$$
$$c_1(\theta) = \max(\tanh(0.5 * \theta_2), \ 0) \ \text{ and } \ c_2(\theta) = \max(\tanh(-0.5 * \theta_2), \ 0).$$

The average loss $L_0$ and individual losses $L_1$ and $L_2$ are shown in Fig. 1. We then pick 5 initial parameter vectors $\theta_{\text{init}} \in \{(-8.5, 7.5), (-8.5, 5), (0, 0), (9, 9), (10, -8)\}$ and plot the corresponding optimization trajectories with different methods in Fig. 3. As shown in Fig. 3, GD gets stuck in 2 out of the 5 runs while other methods all converge to the Pareto set. MGDA and PCGrad converge to different Pareto-stationary points depending on $\theta_{\text{init}}$. CAGrad with $c = 0$ recovers GD and CAGrad with $c = 10$ approximates MGDA well (in theory it requires $c \to \infty$ to exactly recover MGDA).

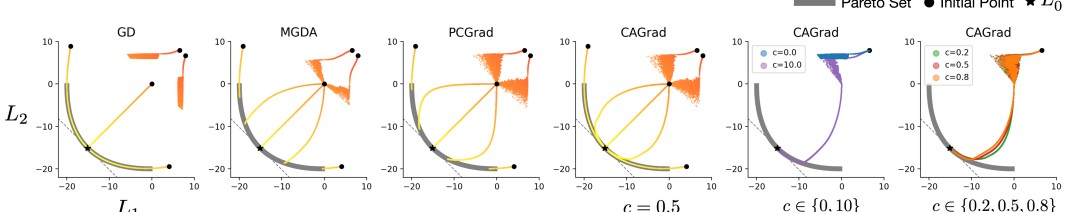

Figure 3: The left four plots are 5 runs of each algorithms from 5 different initial parameter vectors, where trajectories are colored from red to yellow. The right two plots are CAGrad's results with a varying $c \in \{0, 0.2, 0.5, 0.8, 10\}$.

Next, we apply the same set of experiments on the multi-task classification benchmark Multi-Fashion+MNIST [19]. This benchmark consists of images that are generated by overlaying an image from FashionMNIST dataset [39] on top of another image from MNIST dataset [5]. The two images are positioned on the top-left and bottom-right separately. We consider a shrinked LeNet as our model, and train it with Adam [16] optimizer with a $0.001$ learning rate for $50$ epochs using a batch size of $256$. Due to the highly non-convex nature of the neural network, we are not able to visualize the entire Pareto set. But we provide the final training losses of different methods over three independent runs in Fig. 4. As shown, CAGrad achieves the lowest average loss with $c = 0.2$. In addition, PCGrad and MGDA focus on optimizing task 1 and task 2 separately. Lastly, CAGrad with $c = 0$ and $c = 10$ roughly recovers the final performance of GD and MGDA. By increasing $c$, the model performance shifts from more GD-like to more MGDA-like, though due to the non-convex nature of neural networks, CAGrad with $0 \leq c < 1$ does not necessarily converge to the exact same point.

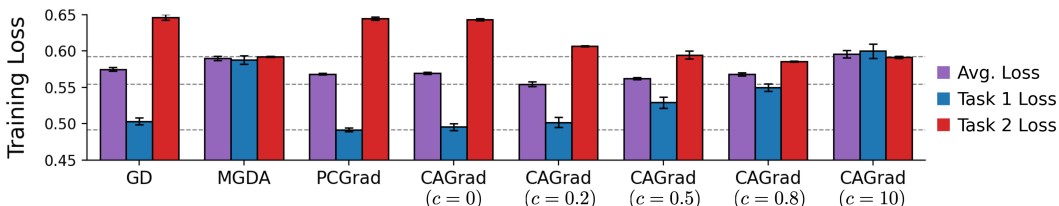

Figure 4: The average and individual training losses on the Fashion-and-MNIST benchmark by running GD, MGDA, PCGrad and CAGrad with different $c$ values. GD gets stuck at the steep valley (the area with a cloud of dots), which other methods can pass. MGDA and PCGrad converge randomly on the Pareto set.

## 5.2 Multi-task Supervised Learning

To answer question (3) in the supervised learning setting, we follow the experiment setup from Yu et al. [41] and consider the NYU-v2 and CityScapes vision datasets. NYU-v2 contains 3 tasks: 13-class semantic segmentation, depth estimation, and surface normal prediction. CityScapes similarly contains 2 tasks: 7-class semantic segmentation and depth estimation. Here, we follow [41] and combine CAGrad with a state-of-the-art MTL method MTAN [21], which applies attention mechanism on top of the SegNet architecture [1]. We compare CAGrad with PCGrad, vanilla MTAN and Cross-Stitch [25], which is another MTL method that modifies the network architecture. MTAN originally experiments with equal loss weighting and two other dynamic loss weighting heuristics [15, 3]. For a fair comparison, all methods are applied under the equal weighting scheme and we use the same training setup from [3]. We search $c \in \{0.1, 0.2, \ldots 0.9\}$ with the best average training loss for CAGrad on both datasets ($0.4$ for NYU-v2 and $0.2$ for Cityscapes). We perform a two-tailed, Student's $t$-test under *equal sample sizes, unequal variance* setup and mark the results that are significant with an $*$. Following Maninis et al.[24], we also compute the average per-task performance drop of method $m$ with respect to the single-tasking baseline $b$: $\Delta m = \frac{1}{K} \sum_{i=1}^{K} (-1)^{l_i} (M_{m,i} - M_{b,i})/M_{b,i}$ where $l_i = 1$ if a higher value is better for a criterion $M_i$ on task $i$ and $0$ otherwise. The single-tasking baseline (independent) refers to training individual tasks with a vanilla SegNet. Results are shown in Tab. 1 and Tab. 2.

Given the single task performance, CAGrad performs better on the task that is overlooked by other methods (Surface Normal in NYU-v2 and Depth in CityScapes) and matches other methods'

| | | Segmentation | | Depth | | Surface Normal | | | | | |
|---|---|---|---|---|---|---|---|---|---|---|---|
| | | (Higher Better) | | (Lower Better) | | Angle Distance (Lower Better) | | Within $t°$ (Higher Better) | | | $\Delta m\% \downarrow$ |
| #P. | Method | mIoU | Pix Acc | Abs Err | Rel Err | Mean | Median | 11.25 | 22.5 | 30 | |
| 3 | Independent | 38.30 | 63.76 | 0.6754 | 0.2780 | 25.01 | 19.21 | 30.14 | 57.20 | 69.15 | |
| $\approx 3$ | Cross-Stitch [25] | 37.42 | 63.51 | 0.5487 | **0.2188** | *28.85 | *24.52 | *22.75 | *46.58 | *59.56 | 6.96 |
| 1.77 | MTAN [21] | 39.29 | 65.33 | 0.5493 | 0.2263 | *28.15 | *23.96 | *22.09 | *47.50 | *61.08 | 5.59 |
| 1.77 | MGDA [30] | *30.47 | *59.90 | *0.6070 | *0.2555 | 24.88 | 19.45 | 29.18 | 56.88 | 69.36 | 1.38 |
| 1.77 | PCGrad [41] | 38.06 | 64.64 | 0.5550 | 0.2325 | *27.41 | *22.80 | *23.86 | *49.83 | *63.14 | 3.97 |
| 1.77 | GradDrop [4] | 39.39 | 65.12 | **0.5455** | 0.2279 | *27.48 | *22.96 | *23.38 | *49.44 | *62.87 | 3.58 |
| 1.77 | CAGrad (ours) | **39.79** | **65.49** | 0.5486 | 0.2250 | 26.31 | 21.58 | 25.61 | 52.36 | 65.58 | **0.20** |

Table 1: Multi-task learning results on NYU-v2 dataset. #P denotes the relative model size compared to the vanilla SegNet. Each experiment is repeated over 3 random seeds and the mean is reported. The best average result among all multi-task methods is marked in bold. MGDA, PCGrad, GradDrop and CAGrad are applied on the MTAN backbone. CAGrad has statistically significant improvement over baselines methods with an ∗, tested with a $p$-value of 0.1.

| | | Segmentation | | Depth | | |
|---|---|---|---|---|---|---|
| | | (Higher Better) | | (Lower Better) | | $\Delta m\% \downarrow$ |
| #P. | Method | mIoU | Pix Acc | Abs Err | Rel Err | |
| 2 | Independent | 74.01 | 93.16 | 0.0125 | 27.77 | |
| $\approx 3$ | Cross-Stitch [25] | *73.08 | *92.79 | *0.0165 | *118.5 | 90.02 |
| 1.77 | MTAN [21] | 75.18 | 93.49 | *0.0155 | *46.77 | 22.60 |
| 1.77 | MGDA [30] | *68.84 | *91.54 | 0.0309 | **33.50** | 44.14 |
| 1.77 | PCGrad [41] | 75.13 | 93.48 | 0.0154 | 42.07 | 18.29 |
| 1.77 | GradDrop [4] | **75.27** | **93.53** | *0.0157 | *47.54 | 23.73 |
| 1.77 | CAGrad (ours) | 75.16 | 93.48 | **0.0141** | 37.60 | **11.64** |

Table 2: Multi-task learning results on CityScapes Challenge. Each experiment is repeated over 3 random seeds and the mean is reported. The best average result among all multi-task methods is marked in bold. PCGrad and CAGrad are applied on the MTAN backbone. CAGrad has statistically significant improvement over baselines methods with an ∗, tested with a $p$-value of 0.1.

performance on the rest of the tasks. We also provide the final test losses and the per-epoch training time of each method in Fig. 5 in Appendix B.2.

### 5.3 Multi-task Reinforcement Learning

To answer question **(3)** in the reinforcement learning (RL) setting, we apply CAGrad on the MT10 and MT50 benchmarks from the Meta-World environment [42]. In particular, MT10 and MT50 contains 10 and 50 robot manipulation tasks. Following [33], we use Soft Actor-Critic (SAC) [10] as the underlying RL training algorithm. We compare against Multi-task SAC (SAC with a shared model), Multi-headed SAC (SAC with a shared backbone and task-specific head), Multi-task SAC + Task Encoder (SAC with a shared model and the input includes a task embedding) [42] and PCGrad [41]. We also compare with Soft Modularization [40] that routes different modules in a shared model to form different policies. Lastly, we also include a recent method (CARE) that considers language metadata and uses a mixture of expert encoder for MTL. We follow the same experiment setup from [33]. The results are shown in Tab. 3. CAGrad outperforms all baselines except for CARE which benefits from extra information from the metadata. We also apply the practical speedup in Sec. 3.3 and sub-sample 4 and 8 tasks for MT10 and MT50 (CAGrad-Fast). CAGrad-fast achieves comparable performance against the state-of-the-art method while achieving a 2x (MT10) and 5x (MT50) speedup over PCGrad. We provide a visualization of tasks from MT10 and MT50, and the comparison of computational efficiency in Appendix B.3.

### 5.4 Semi-supervised Learning with Auxiliary Tasks

Training with auxiliary tasks to improve the performance of a main task is another popular application of MTL. Here, we take semi-supervised learning as an instance. We combine different optimization

| Method | Metaworld MT10 success (mean ± stderr) | Metaworld MT50 success (mean ± stderr) |
|---|---|---|
| Multi-task SAC [42] | 0.49 ±0.073 | 0.36 ±0.013 |
| Multi-task SAC + Task Encoder [42] | 0.54 ±0.047 | 0.40 ±0.024 |
| Multi-headed SAC [42] | 0.61 ±0.036 | 0.45 ±0.064 |
| PCGrad [41] | 0.72 ±0.022 | 0.50 ±0.017 |
| Soft Modularization [40] | 0.73 ±0.043 | 0.50 ±0.035 |
| CAGrad (ours) | **0.83** ±0.045 | **0.52** ±0.023 |
| CAGrad-Fast (ours) | 0.82 ±0.039 | 0.50 ±0.016 |
| CARE [33] | 0.84 ±0.051 | 0.54 ±0.031 |
| One SAC agent per task (upper bound) | 0.90 ±0.032 | 0.74 ±0.041 |

Table 3: Multi-task reinforcement learning results on the Metaworld benchmarks. Results are averaged over 10 independent runs and the best result is marked in bold.

algorithms with Auxiliary Task Reweighting for Minimum-data Learning (ARML) [31], a state-of-the-art semi-supervised learning algorithm. The loss function is composed of the main task and two auxiliary tasks:

$$L_0 = L_{CE}(\theta; D_l) + w_1 L_{aux}^1(\theta; D_u) + w_2 L_{aux}^2(\theta; D_u), \qquad (5)$$

where $L_{CE}$ is the main cross-entropy classification loss on the labeled dataset $D_l$, and $L_{aux}^1, L_{aux}^2$ are auxiliary unsupervised learning losses on the unlabeled dataset $D_u$. We use the same $w_1$ and $w_2$ from ARML, and use the CIFAR10 dataset [17], which contains 50,000 training images and 10,000 test images. 10% of the training images is held out as the validation set. We test PCGrad, MGDA and CAGrad with 500, 1000 and 2000 labeled images. The rest of the training set is used for auxiliary tasks. For all the methods, we use the same labeled dataset, the same learning rate and train them for 200 epochs with the Adam [16] optimizer. Please refer to Appendix B.4 for more experimental details. Results are shown in Tab. 4. With all the different number of labels, CAGrad yields the best averaged test accuracy. We observed that MGDA performs much worse than the ARML baseline, because it significantly overlooks the main classification task. We also compare different gradient manipulation methods on the same task with GradNorm [3], which dynamically adjusts $w_1$ and $w_2$ during training. The results and conclusions are similar to those for ARML.

| Method | 500 labels | 1000 labels | 2000 labels |
|---|---|---|---|
| ARML [31] | 67.05 ±0.16 | 73.22 ±0.26 | 81.35 ±0.36 |
| ARML + PCGrad [41] | 67.49 ±0.64 | 73.23 ±0.62 | 81.91 ±0.19 |
| ARML + MGDA [30] | 49.27 ±0.68 | 60.11 ±2.35 | 60.78 ±0.17 |
| ARML + CAGrad (Ours) | **68.25** ±0.37 | **74.37** ±0.42 | **82.81** ±0.48 |
| GradNorm [3] | 67.35 ±0.15 | 73.53 ±0.23 | 81.03 ±0.71 |
| GradNorm + PCGrad [41] | **67.83** ±0.19 | 73.91 ±0.09 | 82.72 ±0.19 |
| GradNorm + MGDA [30] | 36.99 ±2.11 | 57.94 ±0.92 | 59.12 ±0.63 |
| GradNorm + CAGrad (Ours) | 67.53 ±0.26 | **74.72** ±0.19 | **83.15** ±0.56 |

Table 4: Semi-supervised Learning with auxiliary tasks on CIFAR10. We report the average test accuracy over 3 independent runs for each method and mark the best result in bold.

## 6 Conclusion

In this work, we introduce the Conflict-Averse Gradient descent (CAGrad) algorithm that explicitly optimizes the minimum decrease rate of any specific task's loss while still provably converging to the optimum of the average loss. CAGrad generalizes the gradient descent and multiple gradient descent algorithm, and demonstrates improved performance across several challenging multi-task learning problems compared to the state-of-the-art methods. While we focus mainly on optimizing the average loss, an interesting future direction is to look at main objectives other than the average loss under the multi-task setting.

## Acknowledgements

The research was conducted in the statistical learning and AI group (SLAI) and the Learning Agents Research Group (LARG) in computer science at UT Austin. SLAI research is supported in part by CAREER-1846421, SenSE-2037267, EAGER-2041327, and Office of Navy Research, and NSF AI Institute for Foundations of Machine Learning (IFML). LARG research is supported in part by NSF (CPS-1739964, IIS-1724157, FAIN-2019844), ONR (N00014-18-2243), ARO (W911NF-19-2-0333), DARPA, Lockheed Martin, GM, Bosch, and UT Austin's Good Systems grand challenge. Peter Stone serves as the Executive Director of Sony AI America and receives financial compensation for this work. The terms of this arrangement have been reviewed and approved by the University of Texas at Austin in accordance with its policy on objectivity in research. Xingchao Liu is supported in part by a funding from BP.

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
