# A   Algorithm Details

In this section, we first formally introduce the Multiple Gradient Descent Algorithm and the Projecting Conflicting Gradients method. Then we provide the full proof of Thm. 3.2.

## A.1   Multiple Gradient Descent Algorithm (MGDA)

The Multiple Gradient Descent Algorithm (MGDA) explicitly optimizes towards a Pareto-optimal point for multiple objectives (See the definition 3.1). It is known that a necessary condition for $\theta$ to be a Pareto-optimal point is that we could find a convex combination of the task gradients at $\theta$ that results in the $0$ vector. Therefore, MGDA proposes to minimize the minimum possible convex combination of task gradients:

$$\min \frac{1}{2} \left\| \sum_{i=1}^{K} w_i g_i \right\|^2, \quad \text{s.t.} \quad \sum_{i=1}^{K} w_i = 1, \text{ and } \forall i, w_i \geq 0. \tag{6}$$

We call this the *dual* objective for MGDA, as the primal objective of MGDA has a close connection to CAGrad's primal objective in Eq. (3). Specifically, the *primal* objective of MGDA is

$$\max_{\|d\| \leq 1} \min_i \langle d, g_i \rangle. \tag{7}$$

To see the primal-dual relationship, denote $g_w = \sum_i w_i g_i$, where $w \in \mathcal{W} \triangleq \{w \in \mathbb{R}^K : \sum_i w_i = 1, w_i \geq 0, \forall i \in [K]\}$. Note that $\min_i \langle g_i, d \rangle = \min_{w \in \mathcal{W}} \langle \sum_i w_i g_i, d \rangle$. The Lagrangian of Eq. (7) is

$$\max_d \min_{\lambda \geq 0, w \in \mathcal{W}} \langle d, g_w \rangle - \frac{\lambda}{2} (\|d\|^2 - 1). \tag{8}$$

Since the problem is a convex programming and the Slater's condition holds when $c > 0$ (On the other hand, if $c = 0$, then it is easy to check that all the results hold trivially), the strong duality holds and we can exchange the min and max:

$$\min_{\lambda \geq 0, w \in \mathcal{W}} \max_d \langle d, g_w \rangle - \frac{\lambda}{2} (\|d\|^2 - 1). \tag{9}$$

The optimal $d^* = g_w / \lambda$ and the resulting primal objective is therefore

$$\min_{\lambda \geq 0, w \in \mathcal{W}} \lambda (\frac{1}{2} \|g_w\|^2 + 1). \tag{10}$$

Here, $\lambda$ corresponds to the constraint $\|d\| \leq 1$. If we fix $\lambda$ to be any constant, then we recover the dual objective in Eq. (6).

**Remark**   Looking at the primal form of MGDA in Eq. (7), the major difference between MGDA and CAGrad is that the new update vector $d$ is searched around the $0$ vector for MGDA and $g_0$ for CAGrad. Therefore, theoretically both MGDA and CAGrad optimizes the worst local update, but MGDA is more conservative and can converge to any point on the Pareto set without explicit control (See Thm. 2 from [6]). This also explains MGDA's behavior in practice that it often learns much slower than other methods.

## A.2   Projecting Conflicting Gradients (PCGrad)

Identifying that a major challenge for multi-task optimization is the conflicting gradient, Yu et al. [41] propose to project each task gradient to the normal plane of others before combining them together to form the final update vector. In the following, we provide the full algorithm of the Projecting Conflicting Gradients (PCGrad):

Fig. 2 provides a visualization of PCGrad's update rule for two-task learning (the 3rd column). Different from MGDA and CAGrad, PCGrad does not have a clear optimization objective at each step, which makes it hard to analyze PCGrad's convergence guarantee in general. In practice, the random ordering to do the projection is particularly important for PCGrad to work well [41], which suggests that the intuition of removing the "conflicting" part of each gradient might not be always correct. For the convergence analysis, Yu et al. establishes the convergence guarantee for PCGrad only under the two-task learning setting. Moreover, PCGrad is only guaranteed to converge to the Pareto set without explicit control over which point it will arrive at (See Thm. A.1 in the following).

---

**Algorithm 2** Projecting Conflicting Gradient Update Rule

---

**Input**: model parameter vector $\theta$ and differentiable loss functions $\{L_i\}_{i=1}^K$.
$g_i \leftarrow \nabla_\theta L_i(\theta)$.
$g_i^{\text{PC}} = g_i, \ \forall i$.
**for** task $i \in [K]$ **do**
    **for** $j \neq i \in [K]$ in random order **do**
        **if** $g_i^{\text{PC}} \cdot g_j < 0$ **then**
            $g_i^{\text{PC}} = g_i^{\text{PC}} - \frac{g_i^{\text{PC}} \cdot g_j}{\|g_j\|^2} g_j$.
        **end if**
    **end for**
**end for**
**Return** the new update vector $d = g^{\text{PC}} = \frac{1}{K} \sum_i g_i^{\text{PC}}$.

---

**Theorem A.1** (Convergence of PCGrad [41]). *Consider two-task learning, assume the loss functions $L_1$ and $L_2$ are convex and differentiable. Suppose the gradient of $L_0 = (L_1 + L_2)/2$ is $H$-Lipschitz with $H > 0$. Then, the PCGrad update rule with step size $t \leq 1/H$ will converge to a Pareto-stationary point.*

### A.3 Conflit-Averse Gradient descent (CAGrad)

We provide the full derivation of CAGrad and the proof for its convergence in this section. Our proof assumes $L_0$ is a general function with gradient $g_0 = \nabla L_0$, that is, it does not have to be the average of $L_i$ as the case we focus on in the main paper.

**Lemma A.2.** *Let $d^*$ be the solution of*

$$\max_{d \in \mathbb{R}^m} \min_{i \in [K]} g_i^\top d \quad s.t. \quad \|g_0 - d\| \leq c \|g_0\|,$$

*where $c \geq 0$, and $g_0, g_1, \ldots, g_K \in \mathbb{R}^m$. Then we have*

$$d^* = g_0 + \frac{c \|g_0\|}{\|g_{w^*}\|} g_{w^*},$$

*where $g_{w^*} = \sum_i w_i^* g_i$ and $w^*$ is the solution of*

$$\min_{w \geq \mathcal{W}} g_w^\top g_0 + c \|g_0\| \|g_w\|, \tag{11}$$

*where $\mathcal{W} = \{w \in \mathbb{R}^K : \ \sum_i w_i = 1, \ w_i \geq 0, \forall i \in [K]\}$. In addition,*

$$\min_i g_i^\top d^* = g_{w^*}^\top g_0 + c \|g_0\| \|g_{w^*}\|. \tag{12}$$

*Proof.* Denote $\phi = c^2 \|g_0\|^2$. Note that $\min_i \langle g_i, d \rangle = \min_{w \in \mathcal{W}} \langle \sum_i w_i g_i, d \rangle$. The Lagrangian of the objective in Eq. (3) is

$$\max_{d \in \mathbb{R}^m} \min_{\lambda \geq 0, w \in \mathcal{W}} g_w^\top d - \frac{\lambda}{2}(\|g_0 - d\|^2 - \phi).$$

Since the problem is a convex programming and the Slater's condition holds when $c > 0$ (On the other hand, if $c = 0$, then it is easy to check that all the results hold trivially), the strong duality holds and we can exchange the min and max:

$$\min_{\lambda \geq 0, w \in \mathcal{W}} \max_{d \in \mathbb{R}^m} g_w^\top d - \frac{\lambda}{2} \|g_0 - d\|^2 + \frac{\lambda \phi}{2}.$$

With $\lambda, w$ fixing, the optimal $d$ is achieved when $d = g_0 + g_w/\lambda$, yielding the following dual problem

$$\min_{w, \lambda \geq 0} g_w^\top (g_0 + g_w/\lambda) - \frac{\lambda}{2} \|g_w/\lambda\|^2 + \frac{\lambda}{2} \phi.$$

This is equivalent to

$$\min_{w, \lambda \geq 0} g_w^\top g_0 + \frac{1}{2\lambda} \|g_w\|^2 + \frac{\lambda \phi}{2}.$$

Optimizing out the $\lambda$ we have

$$\min_{w \in \mathcal{W}} g_w^\top g_0 + \sqrt{\phi} \|g_w\|,$$

where the optimal $\lambda = \|g_w\| / \phi^{1/2}$. This solves the problem. (12) is the consequence of the strong duality. □

### Convergence Analysis

**Assumption A.3.** *Assume individual loss functions $L_0, L_1, \ldots, L_K$ are differentiable on $\mathbb{R}^m$ and their gradients $\nabla L_i(\theta)$ are all $H$-Lipschitz, i.e. $\|\nabla L_i(x) - \nabla L_i(y)\| \leq H \|x - y\|$ for $i = 0, 1, \ldots, K$, where $H \in (0, \infty)$. Assume $L_0^* = \inf_{\theta \in \mathbb{R}^m} L_0(\theta) > -\infty$.*

**Theorem A.4** (Convergence of CAGrad). *Assume Assumption A.3 holds. With a fixed step size $\alpha$ satisfying $0 < \alpha \leq 1/H$, we have for the CAGrad in Alg. 1:*

*1) If $0 \leq c < 1$, then CAGrad converges to stationary points of $L_0$ convergence rate in that*

$$\sum_{t=0}^{T} \|g_0(\theta_t)\|^2 \leq \frac{2(L_0(\theta_0) - L_0^*)}{\alpha(1 - c^2)}.$$

*2) For any $c \geq 0$, all the fixed point of CAGrad are Pareto-stationary points of $(L_0, L_1, \ldots, L_K)$.*

*Proof.* We will first prove 1). Consider the $t$-th optimization step and denote $d^*(\theta_t)$ the update direction obtained by solving (3) at the $t$-th iteration. Then we have

$$
\begin{aligned}
L_0(\theta_{t+1}) - L_0(\theta_t) &= L_0(\theta_t - \alpha d^*(\theta_t)) - L_0(\theta_t) \\
&\leq -\alpha g_0(\theta_t)^\top d^*(\theta_t) + \frac{H\alpha^2}{2} \|d^*(\theta_t)\|^2 \\
&\leq -\alpha g_0(\theta_t)^\top d^*(\theta_t) + \frac{\alpha}{2} \|d^*(\theta_t)\|^2 \qquad \textcolor{magenta}{//\alpha \leq 1/H} \\
&\leq -\frac{\alpha}{2} \left( \|g_0(\theta_t\|^2 + \|d^*(\theta_t)\|^2 - \|g_0(\theta_t) - d^*(\theta_t)\|^2 \right) + \frac{\alpha}{2} \|d^*(\theta_t)\|^2 \\
&= -\frac{\alpha}{2} \left( \|g_0(\theta_t)\|^2 - \|d^*(\theta_t) - g_0(\theta_t)\|^2 \right) \\
&\leq -\frac{\alpha}{2}(1 - c^2) \|g_0(\theta_t)\|^2 \qquad \textcolor{magenta}{//\text{by the constraint in (3)}}
\end{aligned}
$$

Using telescoping sums, we have $L_0(\theta_{T+1}) - L_0(0) = -(\alpha/2)(1 - c^2) \sum_{t=0}^{T} \|g_0(\theta_t)\|^2$. Therefore

$$\min_{t \leq T} \|g_0(\theta_t)\|^2 \leq \frac{1}{T+1} \sum_{t=0}^{T} \|g_0(\theta_t)\|^2 \leq \frac{2(L_0(0) - L_0(\theta_{T+1}))}{\alpha(1 - c^2)(T+1)}.$$

Therefore, if $L_0$ is lower bounded, that is, $L_0^* := \inf_{\theta \in \mathbb{R}^m} L_0(\theta) > -\infty$, then $\min_{t \leq T} \|g_0(\theta_t)\|^2 = O(1/T)$.

For general $c \geq 0$, in the fixed point, we have $d^*(\theta) = g_0(\theta) + \lambda g_{w^*}(\theta) = 0$, which readily match the definition of Pareto Stationarity. □

In the following, we show an additional result that when $c \geq 1$, and we use a properly decaying step size, the limit points of CAGrad are either stationary points of $L_0$, or Pareto-stationary points of $(L_1, \ldots, L_K)$.

**Theorem A.5.** *Under Assumption A.3, assume $c \geq 1$ and we a time varying step size satisfying*

$$\alpha_t \leq \frac{\|g_{w_t^*}(\theta_t)\|}{H(c-1) \|g_0(\theta_t)\|},$$

*where $w_t^*$ is the solution of* (11) *at the t-th iteration, then we have*

$$\sum_{t=0}^{T} \alpha_t \left\| g_0(\theta_t) \right\| \left\| g_{w_t^*}(\theta_t) \right\| \leq 2 \frac{\min_i (L_i(\theta_0) - L_i(\theta_{T+1}))}{(c-1)}.$$

Therefore, if we hae $L_i^* = \inf_{\theta \in \mathbb{R}^m} L(\theta) > -\infty$ and $c > 1$, then we have $\alpha_t \left\| g_0(\theta_t) \right\| \left\| g_{w_t^*}(\theta_t) \right\| \to 0$ as $t \to \infty$, meaning that we have either $\alpha_t \to 0$, or $\left\| g_0(\theta_t) \right\| \to 0$ or $\left\| g_{w_t^*}(\theta_t) \right\| \to 0$.

In this case, the actual behavior of the algorithm depends on the specific choice of the step size. For example, if we take $\alpha_t = \frac{\left\| g_{w_t^*}(\theta_t) \right\|}{H(c-1) \left\| g_0(\theta_t) \right\|}$, then the result becomes

$$\sum_{t=0}^{T} \left\| g_{w_t^*}(\theta_t) \right\|^2 \leq 2H \min_i (L_i(\theta_0) - L_i(\theta_{T+1})).$$

which ensures $\left\| g_{w_t^*}(\theta_t) \right\|^2 \to 0$.

*Proof.* For any task $i \in [K]$,

$$L_i(\theta_{t+1}) - L_i(\theta) \leq -\alpha_t g_i(\theta_t)^\top d^*(\theta_t) + \frac{H\alpha_t^2}{2} \left\| d^*(\theta_t) \right\|^2$$

$$\leq -\alpha_t \min_i g_i(\theta_t)^\top d^*(\theta_t) + \frac{H\alpha_t^2}{2} \left\| d^*(\theta_t) \right\|^2$$

$$\leq -\alpha_t \left( g_{w_t^*}(\theta_t)^\top g_0(\theta_t) + c \left\| g_0(\theta_t) \right\| \left\| g_{w_t^*}(\theta_t) \right\| \right) + \frac{H\alpha_t^2}{2} \left\| d^*(\theta_t) \right\|^2 \qquad \textcolor{magenta}{//\text{by (12)}}$$

Meanwhile, note that

$$\left\| d^*(\theta_t) \right\|^2 = \left\| g_0(\theta_t) + \frac{c \left\| g_0(\theta_t) \right\|}{\left\| g_{w_t^*}(\theta_t) \right\|} g_{w_t^*}(\theta_t) \right\|^2$$

$$= (c^2 + 1) \left\| g_0(\theta_t) \right\|^2 + 2 \frac{c \left\| g_0(\theta_t) \right\|}{\left\| g_{w_t^*}(\theta_t) \right\|} g_0(\theta_t)^\top g_{w_t^*}(\theta_t)$$

$$= 2c \frac{\left\| g_0(\theta_t) \right\|}{\left\| g_{w_t^*}(\theta_t) \right\|} \left( g_{w_t^*}(\theta_t)^\top g_0(\theta_t) + c \left\| g_0(\theta_t) \right\| \left\| g_{w_t^*}(\theta_t) \right\| \right) + (1 - c^2) \left\| g_0(\theta_t) \right\|^2.$$

Therefore,

$$L_i(\theta_{t+1}) - L_i(\theta)$$

$$\leq -\alpha_t \left( 1 - H\alpha_t c \frac{\left\| g_0(\theta_t) \right\|}{\left\| g_{w_t^*}(\theta_t) \right\|} \right) \left( g_{w_t^*}(\theta_t)^\top g_0(\theta_t) + c \left\| g_0(\theta_t) \right\| \left\| g_{w_t^*}(\theta_t) \right\| \right) + \frac{H\alpha_t^2}{2} (c^2 - 1) \left\| g_0(\theta_t) \right\|^2$$

$$\overset{(*)}{\leq} -\alpha_t \left( 1 - H\alpha_t c \frac{\left\| g_0(\theta_t) \right\|}{\left\| g_{w_t^*}(\theta_t) \right\|} \right) (c - 1) \left\| g_0(\theta_t) \right\| \left\| g_{w_t^*}(\theta_t) \right\| - \frac{H\alpha_t^2}{2} (c^2 - 1) \left\| g_0(\theta_t) \right\|^2$$

$$= -\alpha_t (c - 1) \left\| g_0(\theta_t) \right\| \left\| g_{w_t^*}(\theta_t) \right\| + \frac{H\alpha_t^2}{2} (c - 1)^2 \left\| g_0(\theta_t) \right\|^2$$

$$\leq -\frac{1}{2} \alpha_t (c - 1) \left\| g_0(\theta_t) \right\| \left\| g_{w_t^*}(\theta_t) \right\| \qquad \textcolor{magenta}{//\text{assume } \alpha_t \leq \frac{\left\| g_{w_t^*}(\theta_t) \right\|}{H(c - 1) \left\| g_0(\theta_t) \right\|}, c \geq 1}$$

where inequality (*) uses Cauchy-Schwarz inequality. Therefore, a telescoping sum gives

$$\sum_{t=0}^{T} \alpha_t \left\| g_0(\theta_t) \right\| \left\| g_{w_t^*}(\theta_t) \right\| \leq 2 \frac{\min_i (L_i(\theta_0) - L_i(\theta_{T+1}))}{(c - 1)},$$

when $c \geq 1$.

$\square$

# B Experiment Details

## B.1 Multi-Fashion+MNIST

**Experiment Details** We follow the experiment setup from [23] and use the same shrinked LeNet that consists of the following layers as the shared base network: CONV(1,5,9,1), MAXPOOL2D(2), RELU, BATCHNORM2D(5), CONV2D(5,10,5,1), MAXPOOL2D(2), RELU, BATCHNORM1D(250), LINEAR(250, 50). Then a task-specific linear head LINEAR(50, 10) is attached to the shared base for the MNIST and FashionMNIST prediction. We use Adam [16] optimizer with a 0.001 learning rate and 0.01 weight decay, and then train for 50 epochs with a batch size of 256. The training set consists of 120000 images of size 36x36 and the test set consists of 20000 images of the same size.

## B.2 Multi-task Supervised Learning

**Experiment Details** For the multi-task supervised learning experiments on the NYU-v2 and CityScapes datasets, we follow exactly the same setup from MTAN [21]. We describe the details in the following. We adopt the SegNet [1] architecture as the backbone network and apply the attention mechanism from MTAN [21] on top of it. For the CityScapes dataset, we use the 7-class semantics labels. We train MTAN, Cross-Stitch, PCGrad and CAGrad with 200 epochs with a batch size of 2 for NYU-v2 and a batch size of 8 for CityScapes, using the Adam [16] optimizer with a learning rate of 0.0001. We further decay the learning rate to 0.00005 at the 100th epoch. As Liu et al. do not separately create a validation set, they average the test performance of each method in the last 10 epochs. We follow this and also average the test performance over the last 10 epochs, but additionally run over 3 seeds and calculate the mean and the standard error. We train CAGrad with $c \in \{0.1, 0.2, 0.3, 0.4, 0.5, 0.6, 0.7, 0.8, 0.9\}$ and pick the best $c$ using their corresponding averaged training performance ($c = 0.4$ for NYU-v2 and $c = 0.4$ for CityScapes).

We also provide the final test losses and the per-epoch training times of each method in Fig. 5.

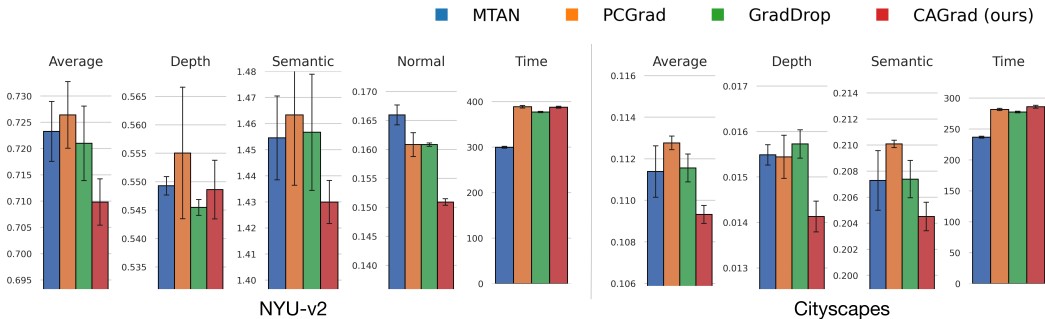

Figure 5: Test loss and training time comparison on NYU-v2 and Cityscapes.

**More Ablation Studies on NYU-v2 and CityScapes Datasets** We conduct the following additional studies on NYU-v2 and CityScapes datasets: 1) How do different methods perform when we additional apply the uncertain weight method [15]? 2) How do CAGrad perform with different values of $c$? 3) How does PCGrad perform when we enlarge the learning rate? Specifically we double the learning rate to 2e-4. Results are provided in Tab. 5 and Tab. 6. We can see that CAGrad perform consistently with different values of $0 < c < 1$. PCGrad with larger learning rate will not perform better. Under the uncertain weights, MTAN and PCGrad indeed perform better but CAGrad is still comparable or better than them.

## B.3 Multi-task Reinforcement Learning

**Experiment Details** The multi-task reinforcement learning experiments follow the exact setup from CARE [33]. Specifically, it is built on top of the MTRL codebase [32]. We consider the MT10 and MT50 benchmarks from the MetaWorld environment [42]. A visualization of the 50 tasks from MT50 is provided in Fig. 6. The MT10 benchmark consists of a subset of 10 tasks from the MT50 task pool. For all methods, we use Soft Actor Critic (SAC) [10] as the underlying reinforcement learning algorithm. All methods are trained over 2 million steps with a batch size of 1280. Following

| | | Segmentation | | Depth | | Surface Normal | | | | | |
|---|---|---|---|---|---|---|---|---|---|---|---|
| #P. | Method | (Higher Better) | | (Lower Better) | | Angle Distance (Lower Better) | | Within $t°$ (Higher Better) | | | $\Delta m\% \downarrow$ |
| | | mIoU | Pix Acc | Abs Err | Rel Err | Mean | Median | 11.25 | 22.5 | 30 | |
| 3 | Independent | 38.30 | 63.76 | 0.6754 | 0.2780 | 25.01 | 19.21 | 30.14 | 57.20 | 69.15 | |
| $\approx$3 | Cross-Stitch [25] | 37.42 | 63.51 | 0.5487 | 0.2188 | 28.85 | 24.52 | 22.75 | 46.58 | 59.56 | 6.96 |
| 1.77 | MTAN [21] | 39.29 | 65.33 | 0.5493 | 0.2263 | 28.15 | 23.96 | 22.09 | 47.50 | 61.08 | 5.59 |
| 1.77 | MGDA [30] | 30.47 | 59.90 | 0.6070 | 0.2555 | 24.88 | 19.45 | 29.18 | 56.88 | 69.36 | 1.38 |
| 1.77 | PCGrad [41] (lr=1e-4) | 38.06 | 64.64 | 0.5550 | 0.2325 | 27.41 | 22.80 | 23.86 | 49.83 | 63.14 | 3.97 |
| 1.77 | PCGrad [41] (lr=2e-4) | 37.70 | 63.40 | 0.5871 | 0.2482 | 28.18 | 24.09 | 21.94 | 47.20 | 60.87 | 8.12 |
| 1.77 | GradDrop [4] | 39.39 | 65.12 | 0.5455 | 0.2279 | 27.48 | 22.96 | 23.38 | 49.44 | 62.87 | 3.58 |
| 1.77 | CAGrad ($c$=0.2) | 39.15 | 65.45 | 0.5563 | 0.2295 | 26.74 | 21.93 | 25.17 | 51.55 | 64.70 | 1.55 |
| 1.77 | CAGrad ($c$=0.4) | 39.79 | 65.49 | 0.5486 | 0.2250 | 26.31 | 21.58 | 25.61 | 52.36 | 65.58 | 0.20 |
| 1.77 | CAGrad ($c$=0.6) | 39.54 | 65.60 | 0.5340 | 0.2199 | 25.87 | 20.94 | 25.88 | 53.78 | 67.00 | -1.36 |
| 1.77 | CAGrad ($c$=0.8) | 39.18 | 64.97 | 0.5379 | 0.2229 | 25.42 | 20.47 | 27.37 | 54.73 | 67.73 | -2.29 |
| 1.77 | MTAN [21] (Uncert. Weights) | 38.74 | 64.70 | 0.5360 | 0.2243 | 26.52 | 21.71 | 25.50 | 52.02 | 65.14 | 0.75 |
| 1.77 | PCGrad [41] (Uncert. Weights) | 37.81 | 64.35 | 0.5318 | 0.2242 | 26.53 | 21.73 | 25.45 | 51.98 | 65.16 | 1.04 |
| 1.77 | CAGrad ($c$=0.2) (Uncert. Weights) | 38.87 | 65.19 | 0.5357 | 0.2227 | 26.38 | 21.64 | 25.66 | 52.21 | 65.39 | 0.319 |
| 1.77 | CAGrad ($c$=0.4) (Uncert. Weights) | 38.89 | 64.98 | 0.5313 | 0.2242 | 25.71 | 20.72 | 26.89 | 54.14 | 67.13 | -1.59 |
| 1.77 | CAGrad ($c$=0.6) (Uncert. Weights) | 39.80 | 65.32 | 0.5334 | 0.2242 | 25.69 | 20.91 | 26.89 | 54.14 | 67.13 | -1.59 |
| 1.77 | CAGrad ($c$=0.8) (Uncert. Weights) | 39.20 | 65.15 | 0.5322 | 0.2202 | 25.28 | 20.17 | 27.83 | 55.41 | 68.25 | -3.14 |

Table 5: Multi-task learning results on NYU-v2 dataset. #P denotes the relative model size compared to the vanilla SegNet. Each experiment is repeated over 3 random seeds and the mean is reported.

| | | Segmentation | | Depth | | |
|---|---|---|---|---|---|---|
| #P. | Method | (Higher Better) | | (Lower Better) | | $\Delta m\% \downarrow$ |
| | | mIoU | Pix Acc | Abs Err | Rel Err | |
| 2 | Independent | 74.01 | 93.16 | 0.0125 | 27.77 | |
| $\approx$3 | Cross-Stitch [25] | 73.08 | 92.79 | 0.0165 | 118.5 | 90.02 |
| 1.77 | MTAN [21] | 75.18 | 93.49 | 0.0155 | 46.77 | 22.60 |
| 1.77 | MGDA [30] | 68.84 | 91.54 | 0.0309 | 33.50 | 44.14 |
| 1.77 | PCGrad [41] | 75.13 | 93.48 | 0.0154 | 42.07 | 18.29 |
| 1.77 | GradDrop [4] | 75.27 | 93.53 | 0.0157 | 47.54 | 23.73 |
| 1.77 | CAGrad ($c$=0.2) | 75.18 | 93.49 | 0.0140 | 40.12 | 13.69 |
| 1.77 | CAGrad ($c$=0.4) | 75.16 | 93.48 | 0.0141 | 37.60 | 11.64 |
| 1.77 | CAGrad ($c$=0.6) | 74.31 | 93.39 | 0.0151 | 34.84 | 11.46 |
| 1.77 | CAGrad ($c$=0.8) | 74.95 | 93.50 | 0.0143 | 36.05 | 10.74 |
| 1.77 | MTAN [21] (Uncert. Weights) | 75.02 | 93.36 | 0.0139 | 35.56 | 9.48 |
| 1.77 | PCGrad [41] (Uncert. Weights) | 74.68 | 93.36 | 0.0135 | 34.00 | 7.26 |
| 1.77 | CAGrad ($c$=0.2) (Uncert. Weights) | 75.05 | 93.45 | 0.0140 | 34.33 | 8.40 |
| 1.77 | CAGrad ($c$=0.4) (Uncert. Weights) | 74.90 | 93.46 | 0.0141 | 34.84 | 9.13 |
| 1.77 | CAGrad ($c$=0.6) (Uncert. Weights) | 74.89 | 93.45 | 0.0136 | 35.17 | 8.48 |
| 1.77 | CAGrad ($c$=0.8) (Uncert. Weights) | 75.38 | 93.48 | 0.0141 | 35.54 | 9.63 |

Table 6: Multi-task learning results on CityScapes Challenge. Each experiment is repeated over 3 random seeds and the mean is reported.

CARE [32], we evaluate each method once every 10000 steps, and report the highest average test performance of a method over 10 random seeds over the entire training stage. For CAGrad-Fast, we sub-sample 4 and 8 tasks randomly at each optimization step as the $S$ (See Eq. (4)) for the MT10 and MT50 experiments. For CAGrad, since MT10 and MT50 have 10 and 50 tasks, much more than the number of tasks in supervised MTL, so instead of using standard optimization library to solve the CAGrad objective, we apply 20 gradient descent steps to approximately solve the objective. The gradient descent is performed with a learning rate of 25 for MT10 and 50 for MT50, with a momentum of 0.5. We search the best $c$ from $\{0.1, 0.5, 0.9\}$ for MT10 and MT50 ($c = 0.9$ for MT10 and $c = 0.5$ for MT50). The computation efficiency is compared in Tab. 7.

In principle, PCGrad should have the same time complexity as CAGrad. However, in practice, PCGrad projects the gradients following a random ordering of the tasks in a sequential fashion (See Alg. 2), so it requires a for loop over that task ordering, which makes it slow for a large number of

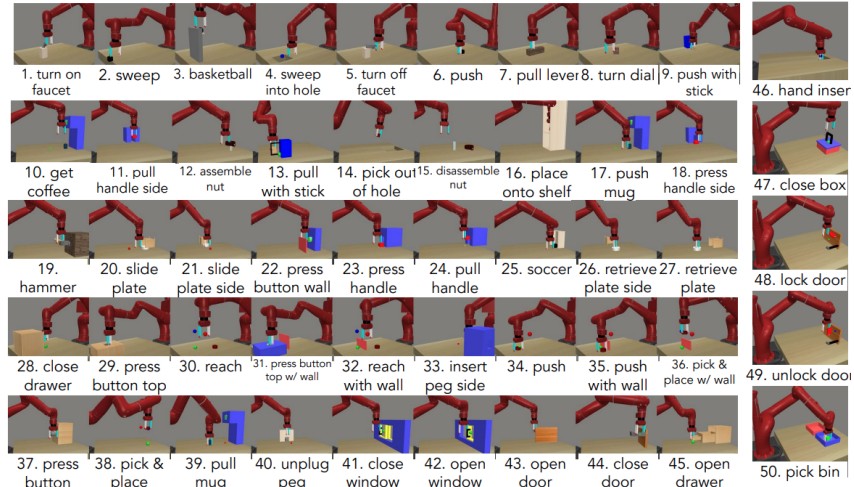

Figure 6: The 50 tasks in MT50 benchmark [42].

| Method | MT10 Time (sec) | MT50 Time (sec) |
|---|---|---|
| PCGrad | 9.7 | 59.8 |
| CAGrad | 10.3 | 27.8 |
| CAGrad-Fast | **4.8** | **11.4** |

Table 7: The training time per update step for PCGrad, CAGrad and CAGrad-Fast on MT10/50.

tasks. Combined with the results from Tab. 3, we see that CAGrad-Fast achieves comparable or better results than PCGrad with a roughly 2x and 5x speedup on MT10 and MT50.

### B.4 Semi-Supervised Learning with Auxiliary Tasks

**Experiment Details** We provide the hyperparameters for reproducing the experiments in our main text. All the methods are applied upon the original ARML baseline, with the same configuration in [31]. Specifically, the batch size is 256 and the optimizer is Adam. The learning rate is initialized to 0.005 in the first 160,000 iterations and decay to 0.001 in the rest iterations. The backbone networks is a `WRN-28-2` model. To stablize the training process, the features are extracted by a moving-averaged model like in [36] with a moving-average factor of 0.95. For PCGrad and MGDA, we use their official implementation without any change. For CAGrad (our method), we fix $c = 0.1$ in all the experiments. The labeled images are randomly selected from the whole training set, and we repeat the experiments for 3 times on the same set of labeled images. We report the test accuracy of the model with the highest validation accuracy.

**Training Losses** We analyze the training losses of different methods to demonstrate the difference between these optimization methods. We report the losses, $L_{CE}$, $L_{aux}^1$ and $L_{aux}^2$, of the last epoch, when the number of labeled images is 2,000. The losses are listed in Tab. 8. We have two key observations: (1) MGDA totally ignores the main task $L_{CE}$, yet it has the smallest loss on the second auxiliary task $L_{aux}^2$. This implies MGDA finds a sub-optimal solution on the Pareto front. (2) PCGrad and CAGrad can both decrease the averaged loss $L_0$ compared with the baseline ARML, however, CAGrad yields a smaller $L_0$ than PCGrad.

| Method | $L_{CE}$ | $L_{aux}^1$ | $L_{aux}^2$ | $L_0$ |
|---|---|---|---|---|
| ARML [31] | **0.0** ±0.0 | 0.0574 ±0.0036 | -0.4946 ±0.0010 | -0.4372 ±0.0046 |
| ARML + PCGrad [41] | **0.0** ±0.0 | 0.0494 ±0.0088 | -0.4943 ±0.0007 | -0.4449 ±0.0095 |
| ARML + MGDA [30] | 0.407 ±0.018 | 0.0453 ±0.0049 | **-0.4980** ±0.0007 | -0.0463 ±0.0233 |
| ARML + CAGrad (Ours) | **0.0** ±0.0 | **0.0419** ±0.0034 | -0.4926 ±0.0023 | **-0.4507** ±0.0058 |

Table 8: The Training Losses in the Last Epoch when the number of the labeled images is $2,000$. Values that are smaller than $10^{-6}$ are replaced by $0$. We report the averaged losses over 3 independent runs for each method, and mark the smallest losses in bold.