# OpenReview forum: "Conflict-Averse Gradient Descent for Multi-task learning"
_NeurIPS.cc/2021/Conference — NeurIPS 2021 Poster_

### Official Review · Reviewer_KEnK · 2021-07-08

**Rating:** 6
**Confidence:** 3

**Summary:**

This work proposes a method to reduce conflicting gradients during training of multi-task learning paradigms via gradient modification. Rather than simply take an average of the per-task gradients, the authors propose to learn a model-wide gradient direction $d$ to update the shared parameters by solving a dual objective to find per-task loss weights. Simply, the proposed method (CAGrad) finds a direction which maximizes the worst local improvement among all tasks, and then applies this gradient update to the shared parameters.

**Limitations And Societal Impact:**

The authors did not discuss the societal impacts of their work. I think they could consider the possible ramifications as done in similar NeurIPS publications on this topic (see [1] and [5]).

[5] Gradient Surgery for Multi-Task Learning (NeurIPS, 2020)


**Main Review:**

This work naturally builds on prior efforts (notably MGDA and MOO) to propose a new algorithm inspired by past work on conflicting gradients (PCGrad) that simply tries to maximize the smallest per-objective change in loss after applying a gradient update (i.e. maximize the worst local improvement among all tasks). The paper is incredibly well-written and easy to read/understand.

It is my opinion that this method is not especially novel or surprising; however, strong empirical findings as well as comprehensive theoretical analysis could more than make up for the lack of novelty. The theoretical results are compelling, while the author's suggested speed-up by sampling a subset of tasks to reduce training-time overhead is also welcome. Nevertheless, there are several areas where the paper could be improved.


-------------
Quite a bit of recent related work is missing from this paper. For instance, recent methods improving upon PCGrad (namely [1] and [2]) are entirely absent.

[1] Just Pick A Sign: Optimizing Deep Multitask Models with Gradient Sign Dropout (NeurIPS, 2020)

[2] Gradient Vaccine: Investigating and Improving Multi-task Optimization in Massively Multilingual Models (ICLR, 2021)

Similarly, the concept of exploring the rate of loss decrease to modify optimization processes has been recently explored in [3] and equation (2) of this paper is remarkably similar to Line 2 of Equation (3) in [4].

[3] Measuring and Harnessing Transference in Multi-Task Learning

[4] Adaptive Auxiliary Task Weighting for Reinforcement Learning

-------------

I would also ask the authors to address the following concerns:

1. **Comparison with prior work.** From Section 5.2, it seems as if Uncertainty Weights are not added to PCGrad to create a "fair comparison"; although PCGrad uses this augmentation for all results reported in their paper.  Given CAGrad solving for $w^{*}$ at every step, thereby dynamically changing the loss weights, how is this fair? Am I missing something?

2. **Loss values as a function of time.** Plotting the train loss as a function of steps to show overfitting/underfitting is not occurring on MultiMNIST for any of the methods presented in Figure 4. Providing the analogous plots for Figure 5 (test loss on NYUv2 and Cityscaps) would also strengthen this paper's empirical findings.

3. **RL Training efficiency.** Similar to point 2, reformulating Table 3 in a similar manner to PCGrad's Figure 3 would be helpful for understanding the relative learning efficiency and convergence of CAGrad.

4. **Hyperparameter tuning discrepancies.** Is the hyperparameter tuning of $c$ done for the experimental results analogous to hyperparameter tuning of the learning rate? Would it be appropriate to tune the learning rate for the other baseline methods in a similar manner (such that the total update magnitude is roughly equivalent, a small grid search over the learning rate)?

5. **Runtime Comparison with PCGrad.** Is the training-time comparison to PCGrad in Figure 5 fair (i.e. using the efficient vectorized_map implementation by the official repo)? How are the quantities so different (600 -- what I assume to be minutes..? compared with just over 400) when both methods take the gradient w.r.t. each task? How does training time compare on MultiFashion/MNIST? What type of hardware is used (see Checklist item 3d)?

For these reasons, I think the experimental results, as presented, are insufficient. I am happy to revise my review during the rebuttal if they can be resolved to strengthen this facet of the paper.

**Time Spent Reviewing:**

2.5

---

> ### Author Response · Authors · 2021-08-10
> **Author Response to Reviewer KEnK**
>
> We thank the reviewer for his/her time and valuable feedback, we address the reviewer's concerns in the following.
>
> **1. It is my opinion that this method is not especially novel or surprising.**
>
> The major novelty of CAGrad is defining a measurement of the “conflict” in multi-task learning and designing an algorithm that explicitly optimizes it with convergence guarantee. While many prior methods also successfully help improve the performance of multi-task learning, most of them are based on heuristics and it is less clear what objective these methods are optimizing for, let alone many of them do not have a general convergence analysis. CAGrad identifies a reasonable objective, for which the algorithm is designed. Moreover, CAGrad provides a unified view of SGD and MGDA.
>
> **2. Quite a bit of recent related work is missing.**
>
> We thank the reviewer for pointing these out and we will include these references into the revision of the paper.
>
> **3. From Section 5.2, it seems as if uncertainty weights are not added to PCGrad to create a fair comparison, though PCGrad uses this augmentation for all results reported in their paper. Given CAGrad solving for $w$ at every step, thereby dynamically changing the loss weights, how is this fair?**
>
> As we pointed out in Figure 2, methods like SGD, MGDA, PCGrad, and CAGrad are all trying to decide a new update d that is some linear combination of the original individual task gradients. This is true also for methods like GradNorm. The difference is that SGD, MGDA and CAGrad compute the weights w by optimizing a specific objective while methods like GradNorm and PCGrad use heuristics. The uncertain weights essentially also reweigh the task gradients. Therefore, all of the above mentioned methods except SGD will have a dynamically changing w, so we maintain that the comparison is fair. We provide the results of applying MTAN, PCGrad, and CAGrad with uncertain weights on NYU-v2 (See NYU-v2-Table-1 from the anonymous link). Results show that CAGrad with uncertain weights still outperforms the baseline methods. Moreover, we believe an ideal MTL method should output the best update from the original task gradients. Stacking multiple different methods together makes it hard to analyze the success.
>
> **4. Plotting the train loss as a function of steps to show overfitting/underfitting is not occurring on MultiMNIST for any of the methods presented in Figure 4. Providing the analogous plots for Figure 5 (test loss on NYUv2 and Cityscaps) would also strengthen this paper's empirical findings.**
>
> We provide the loss curves on multi-mnist and NYU-v2 in the anonymous link (See multi-mnist-train/test-curve and NYU-v2-train/test-curve). We can see that overfitting is indeed happening for NYU-v2 (we have already applied the data augmentation suggested by MTAN to CAGrad and all baseline methods, which PCGrad did not do in their original paper), but the loss curves still show that CAGrad performs better than the baseline methods.
>
> **5. RL Training efficiency. Similar to point 2, reformulating Table 3 in a similar manner to PCGrad's Figure 3 would be helpful for understanding the relative learning efficiency and convergence of CAGrad.**
>
> For the RL experiments on metaworld, we use the numbers reported from Table 1 and Table 3 in Multi-Task Reinforcement Learning with Context-based Representations [27]. And we use Facebook Research’s official implementation (https://github.com/facebookresearch/mtrl) and keep everything the same but just incorporate CAGrad into the implementation. As running any methods on the metaworld benchmarks take days and the reported numbers are across 10 independent runs, we do not have a figure that is similar to PCGrad’s Figure 3. But we provide the learning curve of CAGrad in MT10 and MT50 from the anonymous link.
>
> **6. Would it be appropriate to tune the learning rate for the other baseline methods?**
>
> We use the default learning rate (1e-4) from the MTAN codebase for all methods. We also provide the result of PCGrad with larger learning rates (e.g. 2e-4 and 4e-4) (See NYU-v2-Table-1). PCGrad with larger learning rates results in worse performance.
>
>
> **7. Runtime Comparison with PCGrad. Is the training-time comparison to PCGrad in Figure 5 fair (i.e. using the efficient vectorized_map implementation by the official repo)? What type of hardware is used (see Checklist item 3d)?**
>
> We use Nvidia V100 GPUs for training and the reported time is the per-epoch training time in seconds. For implementation, we use Facebook Research’s official implementation of PCGrad in PyTorch since all experiments’ codebase are written in PyTorch. The reason behind the slowness of PCGrad on the NYU-v2 task is that the implementation of PCGrad requires first calculating a random ordering of tasks, then iteratively calculating the inner products (See https://github.com/facebookresearch/mtrl/blob/main/mtrl/agent/pcgrad.py#L159 and line 3-7 of Algorithm 1 from the PCGrad paper). In the official tensorflow implementation, it should be ​​https://github.com/tianheyu927/PCGrad/blob/master/PCGrad_tf.py#L45. The PCGrad paper mentions that the random ordering helps improve the performance. By contrast, in CAGrad we are not sensitive to the ordering and the gradient inner products between task $i$ and task $j$ for all $i, j$ can be calculated using a single matrix multiplication.

---

> > ### Comment · Reviewer_KEnK · 2021-08-19
> > **Reviewer Response to Authors**
> >
> > Thank you for including these results; I've increased my score from a 5 to a 6.
> >
> > My reasoning is below:
> >
> > The method is intuitive (i.e. find a gradient update that most increases the minimum change in loss for all tasks rather than optimizing the average loss), experimental results appear significant and expressive, and the paper is well written.
> >
> > That said, I maintain my stance with respect to the novelty, impact, and relevancy of this work.
> >
> > Sener's Multi-Task Learning as Multi-Objective Optimization sparked tremendous interest in transferring multi-objective optimization methods to the multi-task deep learning domain. Yu's Gradient Surgery for Multi-Task Learning introduced the concept of directly modifying the gradient direction and magnitude (outside of loss weights) which led to follow-up improvements like GradDrop and GradVac.
> >
> > This work continues to build on the ideas presented by Yu and Sener, but presents a small, but insightful modification which entails significant compute to achieve slightly better empirical results. I would also contest a major novelty of CAGrad relates to the measurement of "conflict" as little/no analysis or comparison is done in this direction (conflict during training?, conflict when parameter capacity is limited?, how does it change as a function of model capacity?, etc.). It seems like this method may improve 'attention focusing' (Ruder's An Overview of Multi-Task Learning in Deep Neural Networks) in addition to or rather than mitigating task conflicts.
> >
> > I have difficulty imagining a follow-up work that builds upon CAGrad -- and while I could be entirely mistaken, wrong, or short-sighted -- it seems to be a terminal node in the chain of research development. Moreover, I would posit the method's computational overhead, the need to search for and set the hyperparameter $c$, and its roughly equivalent performance to PCGrad, GradDrop, GradVac will significantly inhibit its application in industry.

---

### Official Review · Reviewer_Z94j · 2021-07-10

**Rating:** 6
**Confidence:** 5

**Summary:**

Authors propose a multitask learning approach to reduce gradient conflict by minimizing harm to the worst-performing task given any gradient update. Proofs are provided for convergence and pareto optimality of the resultant weight configuration. Experiments are run on a variety of settings, including computer vision and multitask reinforcement learning, with good results throughout.


**Limitations And Societal Impact:**

Yes

**Main Review:**

Overall, I enjoyed reading this paper and find the analysis to be exciting. The results themselves are rather marginal/middling, but given that these are complicated models and multitask learning is known to be quite a messy, high-variance problem setting, I'm happy to be a bit lenient on that. I am glad to see the mathematical analysis and general intuition behind the method, although I have some concerns regarding various elements (including motivation, robustness, and compute overhead). As a lot of this is more clarifications on the method and presumably addressable within the author response, I think the ideas within the paper could be worthwhile to publish at NeurIPS.

(1) My biggest issue with the idea is that I suspect converging to an optimum of the average loss is not necessarily a desideratum. Indeed, given that there is so much literature around balancing loss function weights (MGDA, Uncertainty Weighting, GradNorm, etc.), I suspect that CAGrad is quite brittle when loss function magnitudes are off-balance. MGDA may not converge to the average loss minimum, but that's a feature not a bug, as MGDA is premised on the idea that relative loss magnitudes need to be retuned. For instance, one could arbitrarily increase weight to one loss function until the network is dominated by that task, and in that case it seems that finding a stationary point of the average loss will be a bad idea. So why should we actually want to fit to the average of arbitrarily weighted loss functions?

(2) Have authors tried some of the loss balance methods (listed previously in (1)) in tandem with CAGrad? At the very least it would be nice to benchmark CAGrad against those balancing methods, which I noticed the authors did not do across their different benchmarks.

(3) It would also be interesting to see a benchmark against Gradient Sign Dropout (https://arxiv.org/abs/2010.06808) which is a stochasticity-based method that appeared at NeurIPS last year and was shown to outperform PCGrad in some settings. This particular method at a very high level sparks some similarities as it also starts with the average gradient an then makes slight adjustments based on where the gradient vectors disagree.

(4) Only a computational efficiency comparison is provided between CAGrad, PCGrad, and MTAN. Notably, both CAGrad and PCGrad require multiple full backpropagation passes, while other methods like MGDA, Gradnorm, etc. mitigate the computational overhead by only requiring the backpropagation back to a downstream activation layer. Could authors provide computational comparisons to those methods as well, as this efficiency is some cause for concern?

(5) Authors sweep their main hyperparameter through 10 values - did they take this into account when calculating statistical significance between CAGrad and other baselines?

(6) Is the primary reason for focusing on the worst-case gradient so that the dual problem formulation would work out? Or is there a more intuitive non-mathematical reason for why this is the right optimization problem to solve?

(7) There is always an interesting disconnect in gradient-based MTL methods where the method itself is optimized on the training set while the experiments are all run on the test set. Could the authors comment on potential drawbacks of overfitting to the average loss in the context of generalizability?

(8) Given that there are quite a large corpus of papers at this point that talk about gradient conflict for MTL, the title of the paper is rather ambiguous. Maybe modifying it to explicitly mention the minimax nature of the proposed method would be more appropriate.

**Post-review**: Thanks to the authors for their thoughtful response. I think it is an interesting paper, and I think I am in agreement with authors that further work on loss scaling could further improve the work. As it is now I will maintain my rating at a 6.


**Time Spent Reviewing:**

3

---

> ### Author Response · Authors · 2021-08-10
> **Author Response to Reviewer Z94j**
>
> We thank the reviewer for his/her time and valuable feedback, we address the reviewer's concerns in the following.
>
> **1. My biggest issue with the idea is that I suspect converging to an optimum of the average loss is not necessarily a desideratum. Why should we actually want to fit to the average of arbitrarily weighted loss functions?**
>
> We agree that setting the right balance among each task loss is important and we think it should be set by the user. The core objective of CAGrad is that once the user fixes a final objective $L_0$ (it could be the average loss or some other weighted average loss), it should provably converge to that optimum. Indeed as we pointed out, investigating the main objective $L_0$ other than the average loss might be an interesting future direction.
>
> **2. Have authors tried some of the loss balance methods in tandem with CAGrad? At the very least it would be nice to benchmark CAGrad against those balancing methods, which I noticed the authors did not do across their different benchmarks.**
>
> We include the results of uncertain balancing in NYU-v2-Table-1 from the anonymous link. We can see that the uncertain balancing improves both PCGrad and MTAN. But CAGrad with uncertain balancing still outperforms them. Note that these balancing methods essentially also reweigh the task gradients. We believe a good MTL algorithm should produce a good update given just the original task gradients.
>
> **3. It would also be interesting to see a benchmark against Gradient Sign Dropout.**
>
> We have the GradDrop result on NYU-v2 (See NYU-v2-Table 1 from the anonymous link). CAGrad outperforms GradDrop on this task. We also point out that GradDrop is mainly designed for tasks that share the same inputs as it is mostly effective when the model shares a common hidden representation across tasks. By contrast, methods like PCGrad and CAGrad are agnostic to the model architecture or task inputs.
>
> **4. Only a computational efficiency comparison is provided between CAGrad, PCGrad, and MTAN. Notably, both CAGrad and PCGrad require multiple full backpropagation passes, while other methods like MGDA, Gradnorm, etc. mitigate the computational overhead by only requiring the backpropagation back to a downstream activation layer.**
>
> In Figure 5 we provide the computation time of MTAN v.s. PCGrad and CAGrad. Note that MTAN only requires one backpropagation pass since it averages the losses first and then backpropagate the gradient. So PCGrad and CAGrad are not too inefficient compared to MTAN and other methods. Moreover, MGDA also requires multiple backpropagation passes since it needs all gradients to compute the update. We believe the reviewer is talking about the MGDA-UB method. But note that although methods like MGDA-UB or GradDrop only calculates the gradients to a common activation layer, they in general require a common model backbone and the inputs being shared across tasks, which is not true for models like MTAN (e.g. it has different attention modules for different tasks) and multi-task problems like the metaworld reinforcement learning tasks. In addition, it is in fact possible to derive a similar CAGrad-UB method. But we want to emphasize that CAGrad is a general methods for multi-task learning (in comparison to multi-objective learning where the inputs are shared).
>
> **5. Authors sweep their main hyperparameter through 10 values - did they take this into account when calculating statistical significance between CAGrad and other baselines?**
>
> We only calculate the statistical significance using the best hyperparameters selected using the training loss. In practice we could tune the value of $c$ if we have a separate validation data (for NYU-v2 and Cityscapes, the common setup does not include a separate validation set and most prior works only run the method once with a randon seed).
>
> **6. Is the primary reason for focusing on the worst-case gradient so that the dual problem formulation would work out? Or is there a more intuitive non-mathematical reason for why this is the right optimization problem to solve?**
>
> The reason why we focus on the worst-case gradient is that we realize the success behind PCGrad and CAGrad is that no individual loss is overly sacrificed for the main objective (the average loss for example). Therefore, we first propose how we define the “conflict” (i.e. the $R$ in Eq. 2) and then design CAGrad as a practical algorithm to minimize the conflict when optimizing the main objective.
>
> **7. There is always an interesting disconnect in gradient-based MTL methods where the method itself is optimized on the training set while the experiments are all run on the test set. Could the authors comment on potential drawbacks of overfitting to the average loss in the context of generalizability?**
>
> The reviewer has pointed out a very interesting question regarding the generalization gap. We are not sure how algorithms like MGDA, PCGrad and CAGrad will result in different generalization behavior. But we also do not see any reason why these algorithms will cause more severe overfitting. It could be a very interesting research direction to study the generalization gap for different MTL algorithms.
>
> **8. Given that there are quite a large corpus of papers at this point that talk about gradient conflict for MTL, the title of the paper is rather ambiguous. Maybe modifying it to explicitly mention the minimax nature of the proposed method would be more appropriate.**
>
> Thanks for your suggestion. In fact, we have spent quite some time on the title but haven’t really found a very satisfying one yet. We will keep thinking about it. Meanwhile, if you have any good title in your mind please feel free to let us know about it.

---

### Official Review · Reviewer_Nd5K · 2021-07-14

**Rating:** 7
**Confidence:** 3

**Summary:**

This work presents a first-order multi-task learning method which uses, in place of the average task gradient, a vector which maximises the minimum local single-task improvement in a neighborhood of the average task gradient.

In multi-task learning, the goal is to find the optimal point achieving low losses across all tasks. However, gradient descent methods commonly used for single tasks in computer vision and reinforcement learning achieve suboptimal performance when applied on the average task loss due to conflicting gradients (single tasks gradients pointing in opposite directions). Recently, several methods have proposed modification to the update direction that alleviate this problem. However, these methods are either heuristics or can converge to any point in the pareto optimal set of the task losses (e.g. MGDA [5]), which might not be a minimum of the average task loss.

The proposed method, called Conflict-Averse Gradient descent (CAGrad), introduces an additional hyperparameter $c \geq 0$ which allows to interpolate between gradient descent ($c = 0$) and MGDA ($c \to \infty$). Furthermore, CAGrad provably converges to a stationary point of the average task loss with the same rate as gradient descent (when $c < 1$) or to a pareto stationary point (when $c \geq 0$). At each step of the method, the update direction is obtained by solving  an optimization problem in $K$ dimensions, where $K$ is the number of tasks, but has an overhead comparable to state of the art multi-task learning methods of the same kind (see last plot of Figure 5). Experiments on several computer vision and reinforcement learning multi-task benchmarks, where the method is used to optimize deep network parameters, show that the method achieves state of the art performance in all the settings.


**Limitations And Societal Impact:**

The authors address some but not all the limitations of the method. They do not discuss enough about the newly introduced $c$ hyperparameter which is absent in related methods (see point 6 of the main review).

The authors write in the checklist that their work does not have a negative societal impact. I agree


**Main Review:**

#### **Originality**
1. The method can be considered a fairly straightforward extension of MGDA [5] and it seems novel.  Related works are appropriately cited and it is very clear how the proposed approach compares to them. However, I am not very familiar with recent progress in multitask learning. Contributions are clearly stated in lines 43-51 but I have doubts on some of the claims (see quality section).

#### **Quality**

2. The authors claim that when $c -> \infty$ the algorithm is equivalent to MGDA [5] (line 107). However, while intuitively minimizing $F(w)$ will correspond to the same minimization problem solved by MGDA, the final update direction will be scaled by $\sqrt{\phi} = c ||g_0||$.

3. The authors write that one of the main advantages of the method is that it has convergence guarantees to the stationary point of the average task loss, which PCGrad and MGDA lack. While this is true for MGDA it is usually not true for PCGrad, which either converges to a stationary point of the average task loss or to a point where the directions of the task gradients cancel out [34], although the latter does not usually occur in practice when taking stochastic gradients.

4. All methods in Tables 1 and 2 achieve largely better performance than what is reported in [34]. Why is that? I think this should be discussed in the paper.

5. The proofs before the convergence analysis section seem correct. Theorem 3.3 in the main is different from the one in the supplementary. I will assume the latter to be the correct one. Theorem 3.3 1)  (convergence to stationary point of average task loss with $c < 1$) is proved correctly while the proof for 2) (all fixed points of the method are pareto stationary) is not clear to me: I could not understand how a fixed-point of the algorithm is a pareto stationary point when $g_0 (\theta) = 0$ (lines 382 and 383). 2) is a fundamental result of the proposed method, without it even if you can show that when $c \geq 1$ the method converges you cannot guarantee for it to converge to a pareto stationary point. Furthermore the convergence with $c \geq 1$ is guaranteed (in the supplementary) with time-varying step sizes not used in the experiments or discussed. In the realistic experiments $c < 1$.

6. The experiments are extensive and show a clear advantage of CAGrad in all settings considered. However, the authors should discuss more about the introduced hyperparameter $c$ and how much it is difficult to tune, as a potential limitation of the approach. Note that PCGrad and MGDA, which are the main contenders, have the same hyperparameters as gradient descent.  The experiments in Section 5.1 only study the final training loss varying $c$ for a single artificial setup and the toy example, while the other experiments select the best $c$ looking at the average training error and the optimal value varies largely among different scenarios. Including sensitivity analysis for $c$  in some of the realistic settings and discussing this limitation in the conclusion could address this issue.

#### **Clarity**

7. The main paper is well written and easy to follow. I really appreciated the toy example in Figures 1 and 3 and the illustration in Figure 2 which visually compare CAGrad with other methods. The supplementary was less clear and contained more typos.

#### **Significance**
8. Studying multi-task learning is important to widen the scope of applicability of machine learning models. This work develops a principled, efficient and effective method which can be applied in combination with deep learning models and optimizers and thus it is important for a large part of the machine learning community.

#### **Final Score**

9. I am giving a 5. The proposed method seems promising but the theoretical justification  is lacking and I have some doubts on the experiments. However, I am not very familiar with the topic and I am willing to increase the score if the authors properly address my concerns in the quality section.


#### **Additional comments**

10. In some parts of the paper, like the abstract and intro it is written that the method converges to a minimum or to optimal points. It would be better to say that it converges to stationary points instead since the single task and the average task losses are non-convex both in the theoretical results and in the experiments.

11. eq. (2) and (3) could be misleading. Going forward in the paper shows that the objective that the authors actually minimize is the LHS of (3) but this could also be said close to line 94.

12. Line 199. The authors could briefly describe what the two tasks are (one is to classify the MNIST image and the other to classify the Fashion mnist one).

13. Figure 4. I think that using the same set of seeds for the algorithms could help to make the point more concrete.

14. An interesting addition would be the comparison/discussion on MGDA-UB, which seems to greatly improve the time complexity of MGDA by using an upper bound on the objective used to obtain the update direction. Can a similar upper bound be derived for CAGrad?

#### **Post Rebuttal**
The authors have addressed very well my concerns in the quality section of the review by conducting experiments which analyse the newly introduced hyperparameter c and clarifying parts of the paper I did not fully grasp before. Thus I am increasing the score from 5 to 7 and the confidence from 2 to 3.

#### **References**

[5] Jean-Antoine Désidéri. Multiple-gradient descent algorithm (mgda) for multiobjective optimization.Comptes Rendus Mathematique, 350(5-6):313–318, 2012.

[25] Ozan Sener and Vladlen Koltun. Multi-task learning as multi-objective optimization. arXiv preprint arXiv:1810.04650, 2018.


[34] Tianhe Yu, Saurabh Kumar, Abhishek Gupta, Sergey Levine, Karol Hausman, and Chelsea Finn. Gradient surgery for multi-task learning.arXiv preprint arXiv:2001.06782, 2020.


**Time Spent Reviewing:**

10

---

> ### Author Response · Authors · 2021-08-10
> **Author Response to Reviewer Nd5K**
>
> We thank the reviewer for his/her time and valuable feedback, we address the reviewer's concerns in the following.
>
> **1. The authors claim that when $c\rightarrow\infty$ the algorithm is equivalent to MGDA [5] (line 107). However, while intuitively minimizing $F(w)$ will correspond to the same minimization problem solved by MGDA, the final update direction will be scaled by $\phi=c||g_0||$.**
>
> Yes, indeed the final update direction will be scaled. When we say they are equivalent, we mean that as $c \rightarrow \infty$, the solution to the dual problems will result in the same decision variables $w$. Given sufficiently small learning rate and large enough $c$ (usually $c >= 10$ will be sufficient), then the algorithm behaves almost the same as MGDA in practice. (See Figure 4 for example).
>
> **2. The authors write that one of the main advantages of the method is that it has convergence guarantees to the stationary point of the average task loss, which PCGrad and MGDA lack. While this is true for MGDA it is usually not true for PCGrad, which either converges to a stationary point of the average task loss or to a point where the directions of the task gradients cancel out [34], although the latter does not usually occur in practice when taking stochastic gradients.**
>
> PCGrad proves that the improvement on the average loss scales with $(1-\cos^2 \phi)||g||^2$. Then it claims that as long as the $\cos\phi > -1$, PCGrad will keep optimizing. But when the model is optimized to a point on the Pareto set, by definition, the directions of the task gradients cancel out (otherwise there exists an update vector that both tasks can be optimized). Hence PCGrad will stop at any point on the Pareto set, which is also shown as an example in Figure 1.
>
> **3. All methods in Tables 1 and 2 achieve largely better performance than what is reported in [34]. Why is that?**
>
> The fact is that we use the official implementation of the MTAN (https://github.com/lorenmt/mtan), and the authors keep updating the repo since MTAN’s publication. We believe that PCGrad [34] does not apply any data augmentation and this is the main reason why all our results are largely better. But as overfitting is very severe on NYU-v2, applying data augmentation has become the default setup when comparing multi-task learning algorithms (as suggested from the github repository).
>
> **4. The proofs before the convergence analysis section seem correct. Theorem 3.3 in the main is different from the one in the supplementary. I will assume the latter to be the correct one. Theorem 3.3 1) (convergence to stationary point of average task loss with c<1) is proved correctly while the proof for 2) (all fixed points of the method are pareto stationary) is not clear to me: I could not understand how a fixed-point of the algorithm is a Pareto stationary point when $g_0(θ)=0$**
>
> The claims in the main and the supplementary materials are equivalent. Note that a stationary point of the average loss L0 is also a Pareto-stationary point (this is because if a point has a zero gradient on an average loss, there is no possible update that can improve all losses since the gradients of the different loss have zero mean and hence conflict with each other).
>
> **5. In line 382-383, Theorem A.4. 2) is a fundamental result of the proposed method, without it even if you can show that when $c\geq 1$ the method converges you cannot guarantee for it to converge to a Pareto stationary point. Furthermore the convergence with $c\geq 1$ is guaranteed (in the supplementary) with time-varying step sizes not used in the experiments or discussed. In the realistic experiments $c<1$.**
>
> Indeed CAGrad by design uses $0 < c < 1$. We analyze $c=0$ and $c > 1$ for analysis and unifying SGD and MGDA into the CAGrad family with different $c$. Although in theory CAGrad requires a time-varying step size for convergence, in practice people often apply more complicated optimizers like Adam for faster learning. Therefore following the setup from previous work, we also apply those more advanced optimizers for empirical results.
>
> **6. The authors should discuss more about the introduced hyperparameter $c$ and how much it is difficult to tune, as a potential limitation of the approach. Including sensitivity analysis for c in some of the realistic settings and discussing this limitation in the conclusion could address this issue.**
>
> We provide the results of applying CAGrad with different c values on NYU-v2 in NYU-v2-Table-1 from the anonymous link. As we can see, $c$ does indeed influence the results, but not very much.
>
> **7. Additional comments:**
> - In abstract/intro, it would be better to say that CAGrad converges to stationary points instead.
> - emphasize the objective is minimizing LHS of (3).
> - In line 199, the authors could briefly describe what the two tasks are.
> - In Figure 4, using the same set of seeds would be better.
> - Can a similar upper bound be derived for CAGrad as in MGDA-UB?
>
> We thank the reviewers for this advice and we will incorporate it all in the revised version. In Figure 4, we do use the same set of seeds (0, 1, and 2). CAGrad can also similarly derive a CAGrad-UB algorithm. But note that this UB trick is only possible when all tasks share the same inputs (so essentially the multi-objective instead of multi-task setting). For example, for the RL problems in the metaworld benchmark, MGDA-UB is not applicable.

---

> > ### Comment · Reviewer_Nd5K · 2021-08-19
> > **On the proof of Theorem 3.3 2)**
> >
> > Thanks for the detailed response.  You addressed very well all my concerns regarding the experiments and I really appreciated the additional experiments. However, I still have some doubts on the theoretical results.
> >
> > In particular in Lines 382-383 of the appendix (proof of point 2 in Theorem 3.3) you write *"in the fixed point, we have  $d^{\star}(\theta)=g_{0}(\theta)+\lambda g_{w^{\star}}(\theta)=0$, which readily match the definition of Pareto Stationarity"*. I do not understand why if  $d^{\star}(\theta) = 0$ (apologies for the typo in the review) then $\theta$ is pareto stationary. To me this would be true only if also $g_{0}(\theta) = 0$ and $g_{w^{\star}}(\theta)=0$ which means that the point is also stationary for the average task loss. Furthermore I do not think that $d^{\star}(\theta) = 0$ guarantees that $g_{0}(\theta) = 0$ and $g_{w^{\star}}(\theta)=0$ since I guess there could be cases where for example $g_{0}(\theta) = - \lambda g_{w^{\star}}(\theta) \neq 0$.

---

> > > ### Author Response · Authors · 2021-08-19
> > > **Author Response to Reviewer Nd5K on the proof of Theorem 3.3 2)**
> > >
> > > Thanks for asking! According to Lemma 3.2, the definition of Pareto-stationary is that
> > >
> > > $$\exists w_i \geq 0, \text{s.t.}   \sum_i w_i = 1, \text{and} \sum_i w_i g_i = 0.$$
> > >
> > > Now assume we have $d^*(\theta) = g_0(\theta) + \lambda g_{w^*}(\theta) = 0$, then consider
> > >
> > > $$\forall i, w_i = \frac{1 + K\lambda w^*_i}{K + K\lambda}.$$
> > >
> > > We can show that these {$w_i$} satisfies the Pareto-stationary definition. Because
> > >
> > > $$ \sum_i w_i = \frac{K + K\lambda}{K + K\lambda} = 1, \text{and} \sum_i w_i g_i = \frac{K}{K+K\lambda} \sum_i (\frac{1}{K} + \lambda w^*_i) g_i = 0.$$
> > >
> > > In addition, we did not mean that $d^*(\theta) = 0$ guarantees that $g_0(\theta) = 0$. Instead, we mean that when $g_0(\theta) = 0$, it implies $\theta$ is not only stationary for the average loss $L_0$ but also Pareto-stationary because we can simply choose $w_i = \frac{1}{K}$ as in Lemma 3.2.

---

> > > > ### Comment · Reviewer_Nd5K · 2021-08-19
> > > > **RE: On the proof of Theorem 3.3 2)**
> > > >
> > > > Yes, that sounds about right. Thanks for the clarification. I will shortly update the review increasing the score.

---

### Official Review · Reviewer_w48w · 2021-07-18

**Rating:** 5
**Confidence:** 5

**Summary:**

Multi-task learning (MTL) is an efficient way of learning multiple tasks efficiently by training all tasks together, amortizing their cost by sharing some parameters across tasks. In order to perform this, a weighted sum of the task losses is usually optimized, which can lead to poor final results due to sub-optimal optimization, an effect produced by the competition between tasks for the shared resources. Specifically, when computing the gradient updates for the shared resources, gradients of different tasks can conflict with each other, cancelling each other out (an effect known as *conflicting gradients*).

This work introduces CAGrad, an algorithm which minimizes the average loss function (the network's loss), while maximizing the worst local improvement across all tasks, provably converging to a local minimum of the average loss. This algorithm results in a generalization of the usual gradient descent, as well as the MGDA algorithm. Empirical results show the effectiveness of the method.

**Limitations And Societal Impact:**

I find the discussion of the limitations of the work rather disappointing. In practice, MGDA is slow when it comes to time complexity (because of solving $min_\omega ||g_w||$). This term also appears in CAGrad, does that mean that CAGrad also suffers the same overhead? Why? (lines 244 to 246 make me think so)

**Main Review:**

**Clarity and significance:** I enjoyed reading this paper. I think the paper is well-structured and well-presented, making it joyful to read. This is also partly because of its significance, as I can see CAGrad being impactful and used by the community, resulting in a strict improvement over MGDA. When it comes to the way CAGrad is described and introduced, while everything said is true, I find it a bit misleading. For me, it was way more interesting and clarifying to read the remark in line 339 of the appendix:
> The major difference between MGDA and CAGrad is that the new update vector d is searched around the 0 vector for MGDA and $g_0$ for CAGrad.

This line and the results in that page made me better understand CAGrad and its connections with MGDA, and I truly believe this should be put into the main text.

**Related work and experimental comparisons:** This is one of the weakest points of the paper. While the paper does a good job at presenting PCGrad and MGDA, it leaves out several works completely related with the topic of the paper:
1.  [GradNorm](http://arxiv.org/abs/1711.02257) is mentioned but never compared with (GradNorm's goal is to normalize gradients, and it scales losses just because the gradient is a linear operation).
2. [IMTL-G](https://openreview.net/forum?id=IMPnRXEWpvr) is another method for gradient scaling which is not even mentioned.
3. [RotoGrad](https://arxiv.org/abs/2103.02631) is a method that focuses on reducing negative transfer by aligning task gradients.
4. [GradDrop](https://proceedings.neurips.cc//paper_files/paper/2020/hash/16002f7a455a94aa4e91cc34ebdb9f2d-Abstract.html) is another gradient conflicting solution, contemporary to PCGrad, which solves the issue by randomly dropping gradient elements based on a sign-purity score.

**Working on the last-shared representation:** Most of these recent works address the inherent scalability issues of modifying the gradients of the shared parameters by simply modifying those of the last shared representation (the output of the backbone). Whilst I am not entirely sure how this would apply in the case of using MTAN rather than a usual hard parameter-sharing architecture, I am surprised that this was not thought as a way of scaling CAGrad.

This overlook is even more surprising since the authors claim to compare with the MGDA version of Sener et. al (e.g. in Table 4) rather than the one originally proposed by Jean-Antoine Désidéri. Both papers are mentioned in the manuscript, and my understanding is that the main contribution of the paper of Sener et. al. was proposing MGDA-**UB** (upper-bound), which showed that MGDA could scale for MTL settings by working on the output of the backbone. In reality, here CAGrad is compared with the original MGDA algorithm by Désidéri.

**Experiments.** While the experiments are well described, I have some questions for the authors:
- Why using a shrunk version of LeNet?
- How did you choose the parameters for the FashionMNIST experiments? I have worked with very similar datasets and with 50 epochs convergence was not guaranteed. I would like to know why there are only 3 independent runs in such a simple experiment?
- Do you apply some kind of loss normalization to ensure that losses have comparable values?
- Regarding this last question, I find Figure 4 really confusing. You attempt to minimize the average of the losses, and so one could expect that the optimum of the average of the losses is achieved when both tasks losses are equal (at the end, "you care the same" about both losses), which is what MGDA achieves. I think this figure would be way more understandable (and therefore the discussion) if you showed the results normalizing each task by their initial loss during training.
- Why did you choose c based on the *training* loss instead of the validation loss? This should not be the case.
- Did you run MTAN on NYUv2 with data augmentation? This was included in the official repository of MTAN.

**Minor comments:**
- Caption on tables should be placed above the table.
- Eq. (2) is a bit confusing, I would rather see the first part written as $min \{\frac{1}{\alpha} (L_i(\theta) - L_i(\theta - \alpha d))\}$.
- MTAN references are wrong in the experiments.
- Figure 3 is hardly readable. I would remove the right-most and increase the size of the rest.

I think this is a great paper with a lot of potential, and I will be happy to increase my score if the problems raised above are addressed.

---

**Update:** While I have few concerns when it comes its technical part, I still remain concerned regarding the way the paper presents existing work. It is missing important baselines (e.g. IMTL-G) and completely omits any reference to the set of methods that work on only the last shared representation of the architecture.

I do not feel this is properly justified after rebuttal. First, if not comparing with it, this line of work should be mentioned as it is relevant. Second, in the response this was justified by the chosen experimental setup, but to me this is upside-down: the experimental setup should be chosen to compare with all relevant methods, and no the other way around.

I am thus lowering my score to 5 as I expected a better justification by the authors and I am afraid that, given the nice quality of the paper, it is likely to get some attention and as a consequence harm relevant (and future) work by omitting any reference to prior approaches.

**Time Spent Reviewing:**

7 hours

---

> ### Author Response · Authors · 2021-08-10
> **Author Response to Reviewer w48w**
>
> We thank the reviewer for his/her time and valuable feedback, we address the reviewer's concerns in the following.
>
> **1. For me, it was way more interesting and clarifying to read the remark in line 339 of the appendix: The major difference between MGDA and CAGrad is that the new update vector d is searched around the 0 vector for MGDA and g0 for CAGrad.**
>
> Thanks for the suggestion and we will make this clear in the revision of the paper.
>
> **2. Related work and experimental comparisons: This is one of the weakest points of the paper. While the paper does a good job at presenting PCGrad and MGDA, it leaves out several works completely related with the topic of the paper:**
> - GradNorm is mentioned but never compared with (GradNorm's goal is to normalize gradients, and it scales losses just because the gradient is a linear operation).
> - IMTL-G is another method for gradient scaling which is not even mentioned.
> - RotoGrad is a method that focuses on reducing negative transfer by aligning task gradients.
> - GradDrop is another gradient conflicting solution, contemporary to PCGrad, which solves the issue by randomly dropping gradient elements based on a sign-purity score.
>
> We will include these references into the revision of the paper. In fact, PCGrad achieves state-of-the-art results, outperforming GradNorm (Chen et al., 2018) and Orthogonal Gradients (CosReg) (Suteu & Guo, 2019) based on their original papers. In addition, GradDrop and RotoGrad are mainly designed for models that have a shared backbone which produces a common latent representation across tasks. This means they are mostly effective when the inputs for different tasks are the same and only the prediction outputs differ. For models like MTAN which has different attention modules for different tasks, or problems like the reinforcement learning tasks on metaworld where different tasks have different state inputs, these methods are harder to apply. By contrast, methods like MGDA, PCGrad and CAGrad are easy to apply independent of the model architecture or task inputs. We did not find any open-source implementation of IMTL-G. We provide the experiment results of GradDrop on NYU-v2 (see NYU-v2-Table-1) and the results of GradNorm on the semi-supervised learning tasks (see semi-supervised-learning-GradNorm). We can see that CAGrad still outperforms these methods.
>
> **3. Can CAGrad also work on the last representation as MGDA-UB ?**
>
> Thanks for noticing this. CAGrad indeed can be directly applied to only the last representation layer. However, as in many multi-task settings (like in the metaworld reinforcement learning tasks), the inputs to the network are also different for each task. Under such settings, the trick in MGDA-UB does not apply. In other words, it is only applicable to multi-objective optimization where the input is shared for all objectives.
>
> **4. Why using a shrunk version of LeNet?**
>
> We adopt the same LeNet architecture following the MultiMNIST setup in EPO-search [1] exactly.
>
> **5. For the FashionMNIST experiment, I have worked with very similar datasets and with 50 epochs convergence was not guaranteed.**
>
> According to the plot multi-mnist-train/test-curve, we see that training for 50 epochs roughly achieves  convergence for all methods and we can tell the difference among them.
>
> **6. Do you apply some kind of loss normalization to ensure that losses have comparable values?**
>
> As both the MNIST and FashionMNIST are 10-way classification tasks, we do not apply any loss normalization technique, which is also the default setup in previous literature like EPO-search [1] and PCGrad.
>
> **7. Regarding this last question, I find Figure 4 really confusing. You attempt to minimize the average of the losses, and so one could expect that the optimum of the average of the losses is achieved when both tasks losses are equal (at the end, "you care the same" about both losses), which is what MGDA achieves.**
>
> Although we are trying to minimize the average loss, the optimal point will not necessarily be a point where both losses are equal. For example: L1=1 and L2=4 is considered better than L1 = L2 = 3. Because the average loss is 2.5 < 3. On the other hand, it is indeed the case that MGDA treats all tasks equally in the sense that it ensures the “improvement on loss” of all tasks should be equal. Therefore, the final convergence point largely depends on the initialization. Consider if we are given 2 tasks and a model, where task 1 is under-optimized and the model has a large gradient for task 1 but the model is near a local optimum of task 2. In this case, even if it is possible to improve task 1 a lot while not affecting the performance of task 2, MGDA will not take that move because the best improvement on task 1 is bounded by its improvement on task 2, since it ensures that the improvement on both tasks should be equal. This often causes  slow improvement of MGDA in practice.
>
> **8. Why did you choose $c$ based on the training loss instead of the validation loss?**
>
> Indeed $c$ should be chosen based on the validation loss. But since we did not split the data to have a separate validation set, we just used the training loss.
>
> **9. Did you run MTAN on NYUv2 with data augmentation? This was included in the official repository of MTAN.**
>
> In fact for all experiments on NYU-v2 and Cityscapes, we applied the data augmentation included in the official repository, which is recommended since learning on NYU-v2 can cause significant overfitting.
>
> **10. Minor comments: a) Caption on tables should be placed above the table. b) Eq. (2) is a bit confusing, I would rather see the first part written as $\min_i \alpha(L_i(\theta)−L_i(\theta−\alpha d))$. c) MTAN references are wrong in the experiments. d) Figure 3 is hardly readable. I would remove the right-most and increase the size of the rest.**
>
> Thanks for the suggestion and we will make these clear in the revision of the paper.
>
> **11. Limitations and social impact. I find the discussion of the limitations of the work rather disappointing. In practice, MGDA is slow when it comes to time complexity (because of solving $\min_\omega ||g_w||$). This term also appears in CAGrad, does that mean that CAGrad also suffers the same overhead?**
>
> We will include a full discussion on the limitations of CAGrad in the next version of the paper. First,  solving the optimization problem in Alg.1 exactly indeed incurs some overhead. The overhead in principle is on the same level as MGDA’s. However, as also mentioned in the MGDA paper, the dual objective is dealing with a decision variable w that scales with the number of tasks. All we need to do is to calculate the matrix of the inner-products of gradients $A = GG^\top$, where $G$ is of size [num_tasks, parameter_size]. The actual optimization is often super fast once we have $A$ and nearly all the computation is on calculating this matrix $A$. In fact, $A$ is also calculated in both MGDA and PCGrad. So in order to consider the gradient information, essentially we sacrifice time for deciding on a better update.
>
> **Reference:**
> [1] Mahapatra, Debabrata, and Vaibhav Rajan. "Multi-task learning with user preferences: Gradient descent with controlled ascent in pareto optimization." International Conference on Machine Learning. PMLR, 2020.

---

> ### Author Response · Authors · 2021-09-03
> **Update Response to Reviewer w48w**
>
> We truly appreciate the reviewer pointing out relevant related work, and apologize if we conveyed a lack of willingness to react in the response. Though NeurIPS does not allow modifying the paper as a part of the rebuttal process, please rest assured that we fully intend to
> incorporate all the reviewers' suggestions in the paper's related work section. Furthermore, we will generate additional results comparing CAGrad with variants of prior methods that only apply to the last layer of the model by similarly applying CAGrad to the last layer
> of the model as done, for example, by MGDA-UB. We appreciate the reviewer pointing out this opportunity and fully agree that it will improve the paper.
>
> Like the reviewer, we are optimistic that this paper has the potential to be quite impactful, and are committed to doing everything necessary to properly place it within the context of related research.
>
> Thanks again for your attention to our work.

---

### Official Review · Reviewer_MG2N · 2021-07-20

**Rating:** 7
**Confidence:** 3

**Summary:**

The paper proposes a new solution to multi-task learning  that should improve in cases with gradients conflict of a task loss and the average loss gradient. The solution optimizes for an update vector that maximizes the worst task loss relative improvement. Experiments show the superiority of the proposed solution over similar prior methods.


**Main Review:**

The paper is very well written and easy to follow.
The proposed solution is sound and seem to overcome the existing optimization difficulties of multi-task learning.
Extensive experiments are carried on, targeting different interesting questions.
Method performance is superior or comparable to existing methods.
The main down side is the optimization for an update vector after each update, however, the authors proposed a dual objective with the number of optimized parameters equal to the number of tasks being learned.  Further an approximation is proposed to reduce the computation cost where random tasks are selected for the optimization. This alternative is comparable in performance to existing methods.

On line 88, first order Taylor approximation is used but no discussion on this choice.
Line 95, a hyperparameter is introduced with no discussion its role. However, in the experiments section it is effect is shown empirically.
The practical speedup  is not well motivated and justified.

**Time Spent Reviewing:**

3

---

> ### Author Response · Authors · 2021-08-10
> **Author Response to Reviewer MG2N**
>
> We thank the reviewer for his/her time and valuable feedback, we address the reviewer's concerns in the following.
>
> **1. On line 88, 1st-order Taylor approximation is used but no discussion on this choice.**
>
> Higher order Taylor approximation requires estimating the Hessian of the parameters, which could be too large for neural networks. Therefore we choose to use the 1st order Taylor approximation since we can easily compute the gradients of each task loss with respect to the network parameters.
>
> **2. Line 95, a hyperparameter is introduced with no discussion of its role. However, in the experiments section its effect is shown empirically.**
>
> We discuss the role of $c$ both theoretically (in the Remark from line 106-114 and in section 3.2) and empirically (Section 5.1). For more results of the role of $c$ on the NYU-v2 experiments, please see NYU-v2-Table-1 from the anonymous link.

---

### Author Response · Authors · 2021-08-10
**Author's General Comment**

We thank all reviewers for their time and valuable feedback. We put the additional experiments in the following anonymous link: https://anonymous.4open.science/r/neurips-2021-rebuttal-3172. We will respond to individual reviewers to address their specific concerns separately.

---

### Decision · Program_Chairs · 2021-09-27

**Decision:**

Accept (Poster)

**Comment:**

The manuscript is proposing a multi-task learning method using multi-objective optimization. The proposed algorithm extends the MGDA line of work by simply changing the objective from finding a descent direction that is consistent with all objectives to finding a descent direction that is consistent with all objectives and closest to the average gradient. The proposed algorithm is later analyzed theoretically with Pareto stationarity and convergence guarantees. The authors provide an extensive empirical study with very promising results. Reviewers all enjoyed reading the paper and appreciated the algorithm. Most of the major issues about the paper are on the set-of-baselines side which the authors addressed successfully with additional experiments during the rebuttal perior. I believe the paper is interesting and warrant acceptance.